# $k$-Mixup Regularization for Deep Learning via Optimal Transport

**Kristjan Greenewald**                                        *kristjan.h.greenewald@ibm.com*
*MIT-IBM Watson AI Lab, IBM Research*

**Anming Gu**                                                          *agu2002@bu.edu*
*Boston University*

**Mikhail Yurochkin**                                        *mikhail.yurochkin@ibm.com*
*MIT-IBM Watson AI Lab, IBM Research*

**Justin Solomon**                                                  *jsolomon@mit.edu*
*Massachusetts Institute of Technology*

**Edward Chien**                                                      *edchien@bu.edu*
*Boston University*

**Reviewed on OpenReview:** *https://openreview.net/forum?id=lOegPKSuO4*

## Abstract

Mixup is a popular regularization technique for training deep neural networks that improves generalization and increases robustness to certain distribution shifts. It perturbs input training data in the direction of other randomly-chosen instances in the training set. To better leverage the structure of the data, we extend mixup in a simple, broadly applicable way to *k-mixup*, which perturbs $k$-batches of training points in the direction of other $k$-batches. The perturbation is done with displacement interpolation, i.e. interpolation under the Wasserstein metric. We demonstrate theoretically and in simulations that $k$-mixup preserves cluster and manifold structures, and we extend theory studying the efficacy of standard mixup to the $k$-mixup case. Our empirical results show that training with $k$-mixup further improves generalization and robustness across several network architectures and benchmark datasets of differing modalities. For the wide variety of real datasets considered, the performance gains of $k$-mixup over standard mixup are similar to or larger than the gains of mixup itself over standard ERM after hyperparameter optimization. In several instances, in fact, $k$-mixup achieves gains in settings where standard mixup has negligible to zero improvement over ERM.

## 1 Introduction

Standard mixup (Zhang et al., 2018) is a data augmentation approach that trains models on weighted averages of random pairs of training points. Averaging weights are typically drawn from a beta distribution $\beta(\alpha, \alpha)$, with parameter $\alpha$ such that the generated training set is *vicinal*, i.e., it does not stray too far from the original dataset. Perturbations generated by mixup may be in the direction of *any* other data point instead of being informed by local distributional structure. As shown in Figure 1, this property is a key weakness of mixup that can lead to poor regularization when distributions are clustered or supported on a manifold. With larger $\alpha$, the procedure can result in averaged training points with incorrect labels in other clusters or in locations that stray far from the data manifold.

To address these issues, we present *k-mixup*, which averages random pairs of *sets* of $k$ samples from the training dataset. This averaging is done using optimal transport, with *displacement interpolation*. The sets

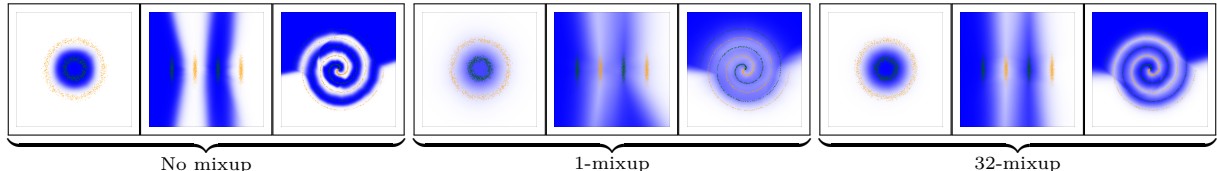

Figure 1: Outputs of a fully-connected network trained on three synthetic datasets for binary classification, with no mixup (ERM), 1-mixup, and 32-mixup regularization ($\alpha = 1$). Note that ERM under-regularizes (visible through the jagged boundaries), 1-mixup over-regularizes (visible through over-smoothing), and that 32-mixup better captures local structure (visible through less blur, increased contrast) while retaining reasonable smoothing between the classes.

of $k$ samples are viewed as discrete distributions and are averaged as distributions in a geometric sense. If $k = 1$, we recover standard mixup regularization. Figures 1 and 2 illustrate how $k$-mixup produces perturbed training datasets that better match the global cluster or manifold support structure of the original training dataset. The constraints of optimal transport are crucial, as for instance a nearest-neighbors approach would avoid the cross-cluster matches necessary for smoothing.[1] Figure 3 illustrates the distribution of possible matchings for a sample point and shows non-zero likelihood for these cross-cluster matches. In Section 5, we provide empirical results that justify the above intuition. The resulting method is easy to implement, computationally cheap, and versatile. Our contributions are as follows.

- Empirical results:
  - We show improved generalization results on standard benchmark datasets showing $k$-mixup with $k > 1$ consistently improves on standard mixup, where $\alpha$ is optimized for both methods. The improvements are consistently similar in magnitude or larger than those of 1-mixup over basic ERM.
  - On image datasets, a heuristic of $k = 16$ outperforms 1-mixup in nearly all cases.
  - We show that $k$-mixup significantly improves robustness to certain distribution shifts (additive noise and adversarial samples) over 1-mixup and ERM.

- Theoretical results:
  - We argue that as $k$ increases, the interpolated samples are more and more likely to remain within the data manifold (Section 3.1).
  - In the clustered setting, we provide an argument that shows inter-cluster regularization interpolates nearest points and better smooths interpolation of labels (Section 3.2).
  - We extend the theoretical analysis of Zhang et al. (2020) and Carratino et al. (2020) to our $k$-mixup setting, showing that it leverages local data distribution structure ($\mathcal{D}_i$ of Eq. 1) to make more informed regularizations (Section 4).

**Related works.** We tackle issues noted in the papers on adaptive mixup (Guo et al., 2019) and manifold mixup (Verma et al., 2018). The first refers to the problem as "manifold intrusion" and seeks to address it by training data point-specific weights $\alpha$ and considering convex combinations of more than 2 points. Manifold mixup deals with the problem by relying on the network to parameterize the data manifold, interpolating in the hidden layers of the network. We show in Section 5 that $k$-mixup can be performed in hidden layers to boost performance of manifold mixup. A related approach is that of GAN-mixup (Sohn et al., 2020), which trains a conditional GAN and uses it to generate data points between different data manifolds. The approaches above require training additional networks and are far more complex than our $k$-mixup method.

---

[1]To see this, note that because nearest-neighbors can be a many-to-one matching, nearly all matches would be intra-cluster between points of the same class and thus provide few/no interpolated labels, particularly missing any interpolations in the voids between classes where interpolation is most important. As additional support, we ran 20 Monte Carlo trials of the CIFAR-10 experiment below with a $k = 16$-nearest-neighbors strategy. It failed to outperform even 1-mixup (0.09% worse). Further discussion is presented in supplement Section J, with an analogue of Figure 3 for a $k$-nearest-neighbors strategy.

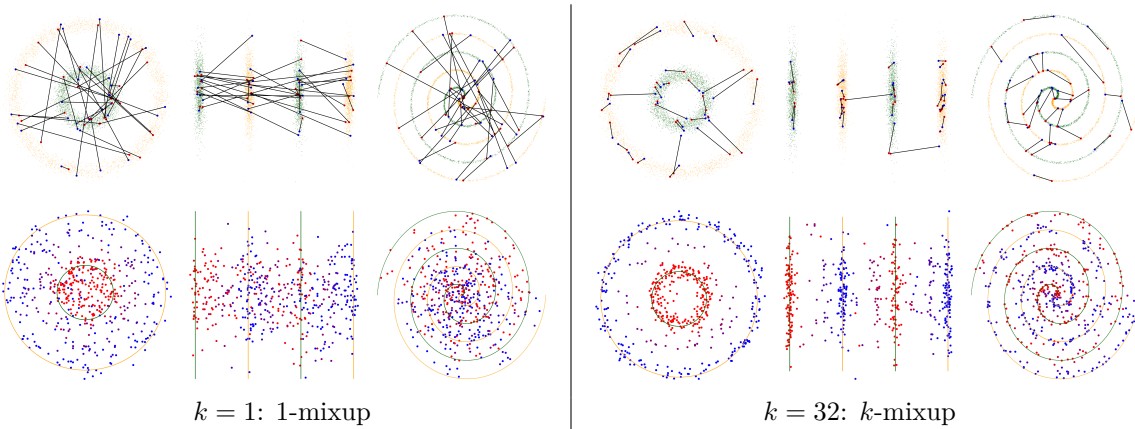

$k = 1$: 1-mixup  $k = 32$: $k$-mixup

Figure 2: Optimal transport couplings and vicinal datasets for $k = 1$ (left) and $k = 32$ (right) in 3 simple datasets. In the bottom row, $\alpha = 1$ was used to generate vicinal datasets of size 512.

The recent local mixup method (Baena et al., 2022) uses a distance-based approach for modifying mixup. This method retains the random matching strategy of mixup, but scales the contribution of a vicinal point to the objective function according to its distance from the original training point. As noted previously, such methods that employ random matching will fail to provide substantive data smoothing for the output function between clusters or high-density regions in the training data.

PuzzleMix (Kim et al., 2020) also combines optimal transport ideas with mixup, extending CutMix (Yun et al., 2019) to combine pairs of images. PuzzleMix uses transport to shift saliency regions of images, producing meaningful combinations of input training data. Their use of optimal transport is fundamentally different from ours and does not generalize to non-image data. There are several other image-domain specific works that are in this vein, including CoMix (Kim et al., 2021), Automix (Liu et al., 2021), and Stackmix (Chen et al., 2021).

Performing optimal transport between empirical samples of distributions has been considered in studies of the *sample complexity* of Wasserstein distances (e.g. Weed & Bach (2019)). Unlike most settings, in our application the underlying source and target distributions are the same; the theoretical investigation of a generalization of variance called $k$-variance by Solomon et al. (2020) considers a similar setting. In other works, transport between empirical samples has been dubbed *minibatch optimal transport* and has been used in generative models (Genevay et al., 2018; Fatras et al., 2020) and domain adaptation (Damodaran et al., 2018; Fatras et al., 2021).

## 2 Generalizing Mixup

**Standard mixup.** Mixup uses a training dataset of feature-target pairs $\{(x_i, y_i)\}_{i=1}^N$, where the target $y_i$ is a one-hot vector for classification. Weighted averages of training points construct a vicinal dataset:

$$(\tilde{x}_{ij}^\lambda, \tilde{y}_{ij}^\lambda) := (\lambda x_i + (1 - \lambda)x_j, \lambda y_i + (1 - \lambda)y_j).$$

$\lambda$ is sampled from a beta distribution, $\beta(\alpha, \alpha)$, with parameter $\alpha > 0$ usually small so that the averages are near an endpoint. Using this vicinal dataset, empirical risk minimization (ERM) becomes:

$$\mathcal{E}_1^{mix}(f) := \mathbb{E}_{i,j,\lambda} \left[ \ell \left( f \left( \tilde{x}_{ij}^\lambda \right), \tilde{y}_{ij}^\lambda \right) \right],$$

where $i, j \sim \mathcal{U}\{1, \ldots, N\}, \lambda \sim \beta(\alpha, \alpha)$, $f$ is a proposed feature-target map, and $\ell$ is a loss function. Effectively, one trains on datasets formed by averaging random pairs of training points. As the training points are randomly selected, this construction makes it likely that the vicinal data points may not reflect the local structure of the dataset, as in the clustered or manifold-support setting.

$k$-**mixup.** To generalize mixup, we sample two random subsets of $k$ training points $\{(x_i^\gamma, y_i^\gamma)\}_{i=1}^k$ and $\{(x_i^\zeta, y_i^\zeta)\}_{i=1}^k$. For compactness, let $x^\gamma := \{x_i^\gamma\}_{i=1}^k$ and $y^\gamma := \{y_i^\gamma\}_{i=1}^k$ denote the feature and target sets

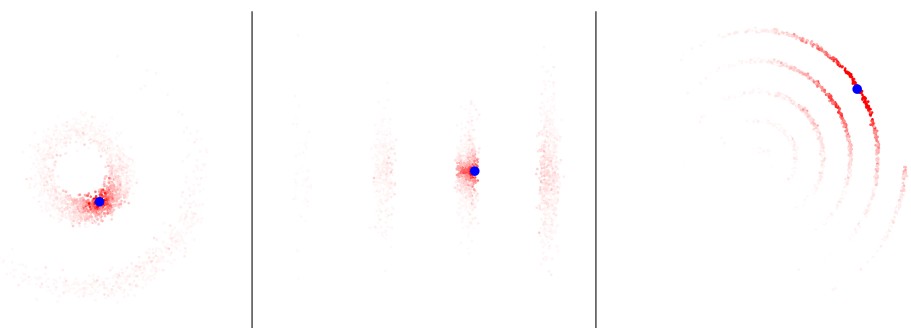

Figure 3: Locally-informed matching distributions $\mathcal{D}_i$ for $k = 32$, for a randomly selected point (see Equation 1 for explicit definition). These distributions reflect local manifold and cluster structure. Note that a nearest-neighbors approach would not provide cross-cluster matchings (see Figure 7 in Appendix J).

(and likewise for $\zeta$). A weighted average of these subsets is formed with *displacement interpolation* and used as a vicinal training set. This concept is from optimal transport (see, e.g., Santambrogio (2015)) and considers $(x^\gamma, y^\gamma)$ and $(x^\zeta, y^\zeta)$ as uniform discrete distributions $\hat{\mu}^\gamma, \hat{\mu}^\zeta$ over their supports. In this setting, the optimal transport problem becomes a linear assignment problem (Peyré & Cuturi, 2019). The optimal map is described by a permutation $\sigma \in S_k$ minimizing the cost:

$$W_2^2(\hat{\mu}^\gamma, \hat{\mu}^\zeta) = \frac{1}{k} \sum_{i=1}^{k} \|x_i^\gamma - x_{\sigma(i)}^\zeta\|_2^2.$$

Here, $\sigma$ can be found efficiently using the Hungarian algorithm (Bertsimas & Tsitsiklis, 1997). Figure 2 gives intuition for this identification. When compared to the random matching used by standard mixup, our pairing is more likely to match nearby points and to make matchings that better respect local structure—especially by having nontrivial probability for cross-cluster matches between nearby points on the two clusters (see Figure 3).

Given a permutation $\sigma$ and weight $\lambda$, the displacement interpolation between $(x^\gamma, y^\gamma)$ and $(x^\zeta, y^\zeta)$ is:

$$DI_\lambda((x^\gamma, y^\gamma), (x^\zeta, y^\zeta)) := \left\{ \lambda(x_i^\gamma, y_i^\gamma) + (1-\lambda)(x_{\sigma(i)}^\zeta, y_{\sigma(i)}^\zeta) \right\}_{i=1}^{k}.$$

As in standard mixup, we draw $\lambda \sim \beta(\alpha, \alpha)$. For the loss function, we consider sampling $k$-subsets of the $N$ samples at random, which we can mathematically describe as choosing $\gamma, \zeta \sim \mathcal{U}\{1, \dots, \binom{N}{k}\}$ for which $\{\{(x_i^\gamma, y_i^\gamma)\}_{i=1}^{k}\}_{\gamma=1}^{\binom{N}{k}}$ are the possible subsets.[2] This yields an expected loss

$$\mathcal{E}_k^{mix}(f) = \mathbb{E}_{\gamma, \zeta, \lambda} \left[ \ell(f(DI_\lambda(x^\gamma, x^\zeta)), DI_\lambda(y^\gamma, y^\zeta)) \right].$$

The localized nature of the matchings makes it more likely that the averaged labels will smoothly interpolate over the decision boundaries. A consequence is that $k$-mixup is robust to higher values of $\alpha$, since it is no longer necessary to keep $\lambda$ close to 0 or 1 to avoid erroneous labels. This can be seen in our empirical results in Section 5 and theoretical analysis in Section 4.

**$k$, $\alpha$, and Root Mean Squared Perturbation Distance.** When $\alpha$ is kept fixed and $k$ is increased, the perturbations become more local, and the distance of matchings (and thus perturbations) decreases. We found that more sensible comparisons across values of $k$ can be obtained by *increasing $\alpha$ in concert with $k$*,[3] so that the average perturbation's squared distance remains constant—bearing in mind that large squared distance values may not be achievable for high values of $k$. In effect, this is viewing the root mean squared perturbation distance as the parameter, instead of $\alpha$. Computation details are found in Section 5.1. Throughout training, this tuned $\alpha$ is kept fixed. We typically pick an $\alpha$ for $k = 1$, get the associated root mean squared distance ($\xi$), and increase $\alpha$ as $k$ increases to maintain the fixed value of $\xi$.

---

[2]These possible subsets need not be enumerated since we optimize the loss with stochastic gradient descent.
[3]Note that this adjustment is of course not necessary for parameter tuning in practice.

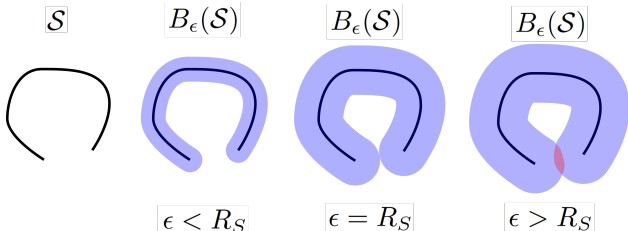

Figure 4: Injectivity radius example. Various $\epsilon$-neighborhoods, $B_\epsilon(\mathcal{S})$, have been illustrated for a curve $\mathcal{S} \subset \mathbb{R}^2$. Intuitively, the topology of $B_\epsilon(\mathcal{S})$ no longer reflects that of $\mathcal{S}$ when $\epsilon > R_S$, the injectivity radius. To be precise, $B_\epsilon(\mathcal{S})$ is no longer homotopy equivalent to $\mathcal{S}$ (see Hatcher (2000) for definition).

**Pseudocode and Computational Complexity.** We emphasize the speed and simplicity of our method. We have included a brief PyTorch pseudocode in Section 5.1 below and note that with CIFAR-10 and $k = 32$, the use of $k$-mixup added about one second per epoch on GPU. Section 5.1 also contains a more extended analysis of computational cost, showing that there is little computational overhead to incorporating $k$-mixup in training compared to ERM alone.

## 3   Manifold & Cluster Structure Preservation

Using an optimal coupling for producing vicinal data points makes it likely that vicinal data points reflect local dataset structure. Below we argue that as $k$ increases, the vicinal couplings will preserve manifold support, preserve cluster structure, and interpolate labels between clusters.

### 3.1   Manifold support

Suppose our training data is drawn from a distribution $\mu$ on a $d$-dimensional embedded submanifold $\mathcal{S}$ in $\mathcal{X} \times \mathcal{Y} \subset \mathbb{R}^M$, where $\mathcal{X}$ and $\mathcal{Y}$ denote feature and target spaces. We define an injectivity radius:

**Definition 3.1** (Injectivity radius). Let $B_\epsilon(\mathcal{S}) = \{p \in \mathcal{X} \times \mathcal{Y} \,|\, \mathsf{d}(p, \mathcal{S}) < \epsilon\}$ denote the $\epsilon$-neighborhood of $\mathcal{S}$ where $\mathsf{d}(p, \mathcal{S})$ is the Euclidean distance from $p$ to $\mathcal{S}$. Define the injectivity radius $R_\mathcal{S}$ of $\mathcal{S}$ to be the infimum of the $\epsilon$'s for which $B_\epsilon(\mathcal{S})$ is not homotopy equivalent to $\mathcal{S}$.

As $\mathcal{S}$ is embedded, $B_\epsilon(\mathcal{S})$ is homotopy equivalent to $\mathcal{S}$ for small enough $\epsilon$, so $R_\mathcal{S} > 0$. Essentially, Definition 3.1 grows an $\epsilon$-neighborhood until the boundary intersects itself. See Figure 4 for a schematic example that illustrates this. We have the following:

**Proposition 3.2.** *For a data distribution $\mu$ supported on an embedded submanifold $\mathcal{S}$ with injectivity radius $R_\mathcal{S}$, with high probability any constant fraction $1 - \delta$ (for any fixed $\delta \in (0, 1]$) of the interpolated samples induced by $k$-mixup will remain within $B_\epsilon(\mathcal{S})$, for $k$ large enough.*

The proof of this proposition is in supplement Section D. Hence, for large enough $k$, the interpolation induced by optimal transport will approximately preserve the manifold support structure. While the theory requires high $k$ to achieve a tight bound, our empirical evaluations show good performance in the small $k$ regime. Some schematic examples are shown in Figures 1, 2, and 3.

### 3.2   Clustered distributions

With a clustered data distribution, we preserve global structure by including many within-cluster and inter-cluster matches, where inter-cluster matches correspond (approximately) to the nearest points across clusters. This contrasts with 1-mixup, which, when the number of clusters is large, is biased towards primarily inter-cluster matches and does not seek to provide any structure to these random matches. These approximate characterizations of $k$-mixup are achieved exactly as $k \to \infty$, as argued below.

To make precise the notion of a clustered distribution, we adopt the $(m, \Delta)$-clusterable definition used by Weed & Bach (2019); Solomon et al. (2020). In particular, a distribution $\mu$ is $(m, \Delta)$-clusterable if $\operatorname{supp}(\mu)$

lies in the union of $m$ balls of radius at most $\Delta$. Now, if our training samples $(x_i, y_i)$ are sampled from such a distribution, where the clusters are sufficiently separated, then optimal transport will prioritize intra-cluster matchings over cross-cluster matchings.

**Lemma 3.3.** *Draw two batches of samples $\{p_i\}_{i=1}^N$ and $\{q_i\}_{i=1}^N$ from a $(m, \Delta)$-clusterable distribution, where the distance between any pair of covering balls is at least $2\Delta$. If $r_i$ and $s_i$ denote the number of samples in cluster $i$ in batch 1 and 2, respectively, then the optimal transport matching will have $\frac{1}{2}\sum_i |r_i - s_i|$ cross-cluster matchings.*

The proof of the above (supplement Section E) involves the pigeonhole principle and basic geometry. We also argue that the fraction of cross-cluster identifications approaches zero as $k \to \infty$ and characterize the rate of decrease. The proof (supplement Section F) follows via Jensen's inequality:

**Theorem 3.4.** *Given the setting of Lemma 3.3, with probability masses $p_1, \ldots, p_m$, and two batches of size $k$ matched with optimal transport, the expected fraction of cross-cluster identifications is $O\left((2k)^{-1/2} \sum_{i=1}^m \sqrt{p_i(1-p_i)}\right)$.*

Note that the absolute number of cross-cluster matches still increases as $k$ increases, providing more information in the voids between clusters. We emphasize that the $(m, \Delta)$ assumption corresponds to a setting where clusters are well-separated such that cross-cluster identifications are as unlikely as possible. Hence, in real datasets, the fraction of cross-cluster identifications will be larger than indicated in Theorem 3.4. Finally, we show that these cross-cluster matches are of length close to the distance between the clusters with high probability (proved in supplement Section G), i.e., the endpoints of the match lie in the parts of each cluster closest to the other cluster. This rests upon the fact that we are considering the $W_2$ cost, for which small improvements to long distance matchings yield large cost reductions in squared Euclidean distance.

**Theorem 3.5.** *Suppose density $p$ has support on disjoint compact sets (clusters) $A, B$ whose boundaries are smooth, where $p > 0$ throughout $A$ and $B$. Let $D$ be the Euclidean distance between $A$ and $B$, and let $R_A, R_B$ be the radii of $A, B$ respectively. Define $A_\epsilon$ to be the subset of set $A$ that is less than $(1 + \epsilon)D$ distance from $B$, and define $B_\epsilon$ similarly. Consider two batches of size $k$ drawn from $p$ and matched with optimal transport. Then, for $k$ large enough, with high probability all cross-cluster matches will have an endpoint each in $A_\epsilon$ and $B_\epsilon$, where $\epsilon = \frac{\max(R_A, R_B)^2}{D^2}$.*

Theorem 3.5 implies that for large $k$, the vicinal distribution created by sampling along the cross-cluster matches will almost entirely lie in the voids between the clusters. If the clusters correspond to different classes, this will directly encourage the learned model to smoothly interpolate between the class labels as one transitions across the void between clusters. This is in contrast to the random matches of 1-mixup, which create vicinal distributions that can span a line crossing any part of the space without regard for intervening clusters. The behaviors noted by Theorems 3.4 and 3.5 are visualized in Figures 1, 2, and 3, showing that $k$-mixup provides smooth interpolation between clusters, and strengthens label preservation within clusters.

## 4 Regularization Expansions

Two recent works analyze the efficacy of 1-mixup perturbatively (Zhang et al., 2020; Carratino et al., 2020). Both consider quadratic Taylor series expansions about the training set or a simple transformation of it, and they characterize the regularization terms that arise in terms of label and Lipschitz smoothing. We adapt these expansions to $k$-mixup and show that the resulting regularization is more locally informed via the optimal transport coupling.

In both works, perturbations are sampled from a globally informed distribution, based upon all other samples in the data distribution. In $k$-mixup, these distributions are defined by the optimal transport couplings. Given a training point $x_i$, we consider all $k$-samplings $\gamma$ that might contain it, and all possible $k$-samplings $\zeta$ that it may couple to. A locally-informed distribution is the following:

$$\mathcal{D}_i := \frac{1}{\binom{N-1}{k-1}\binom{N}{k}} \sum_{\gamma=1}^{\binom{N-1}{k-1}} \sum_{\zeta=1}^{\binom{N}{k}} \delta_{\sigma_{\gamma\zeta}(x_i)}, \tag{1}$$

where $\sigma_{\gamma\zeta}$ denotes the optimal coupling between $k$-samplings $\gamma$ and $\zeta$. This distribution will be more heavily weighted on points that $x_i$ is often matched with. An illustration of this distribution for a randomly selected point in our synthetic examples is visible in Figure 3. We use "locally-informed" in the sense of upweighting of points closer to the point of interest that are likely to be matched to it by $k$-mixup.

Zhang et al. (2020) expand about the features in the training dataset $\mathcal{D}_X := \{x_1, \ldots, x_n\}$, and the perturbations in the regularization terms are sampled from $\mathcal{D}$. We generalize their characterization to $k$-mixup, with $\mathcal{D}$ replaced by $\mathcal{D}_i$. Focusing on the binary classification problem for simplicity, we assume a loss of the form $\ell(f(x), y) = h(f(x)) - y \cdot f(x)$ for some twice differentiable $h$ and $f$. This broad class of losses includes the cross-entropy for neural networks and all losses arising from generalized linear models.

**Theorem 4.1.** *Assuming a loss $\ell$ as above, the $k$-mixup loss can be written as:*

$$\mathcal{E}_k^{mix}(f) = \mathcal{E}^{std} + \sum_{j=1}^{3} \mathcal{R}_j + \mathbb{E}_{\lambda \sim \beta(\alpha+1, \alpha)}[(1-\lambda)^2 \phi(1-\lambda)],$$

*where $\lim_{a \to 0} \phi(a) = 0$, $\mathcal{E}^{std}$ denotes the standard ERM loss, and the three $\mathcal{R}_i$ regularization terms are:*

$$\mathcal{R}_1 = \frac{\mathbb{E}_{\lambda \sim \beta(\alpha+1,\alpha)}[1-\lambda]}{n} \times \sum_{i=1}^{N} (h'(f(x_i)) - y_i) \nabla f(x_i)^T \mathbb{E}_{r \sim \mathcal{D}_i}[r - x_i]$$

$$\mathcal{R}_2 = \frac{\mathbb{E}_{\lambda \sim \beta(\alpha+1,\alpha)}[(1-\lambda)^2]}{2n} \times \sum_{i=1}^{N} h''(f(x_i)) \nabla f(x_i)^T \mathbb{E}_{r \sim \mathcal{D}_i}[(r - x_i)(r - x_i)^T] \nabla f(x_i)$$

$$\mathcal{R}_3 = \frac{\mathbb{E}_{\lambda \sim \beta(\alpha+1,\alpha)}[(1-\lambda)^2]}{2n} \times \sum_{i=1}^{N} (h'(f(x_i)) - y_i) \mathbb{E}_{r \sim \mathcal{D}_i}[(r - x_i) \nabla^2 f(x_i)(r - x_i)^T].$$

A proof is given in Section H of the supplement and follows from some algebraic rearrangement and a Taylor expansion in terms of $1 - \lambda$. The higher-order terms are captured by $\mathbb{E}_{\lambda \sim \beta(\alpha+1,\alpha)}[(1-\lambda)^2 \phi(1-\lambda)]$. $\mathcal{E}^{std}$ represents the constant term in this expansion, while the regularization terms $\mathcal{R}_i$ represent the linear and quadratic terms. These effectively regularize $\nabla f(x_i)$ and $\nabla^2 f(x_i)$ with respect to local perturbations $r - x_i$ sampled from $\mathcal{D}_i$, ensuring that our regularizations are locally-informed. In other words, the regularization terms vary over the support of the dataset, at each point penalizing the characteristics of the locally-informed distribution rather than those of the global distribution. This allows the regularization to adapt better to local data (e.g. manifold) structure. For example, $\mathcal{R}_2$ and $\mathcal{R}_3$ penalize having large gradients and Hessians respectively along the directions of significant variance of the distribution $\mathcal{D}_i$ of points $x_i$ is likely to be matched to. When $k = 1$, this $\mathcal{D}_i$ will not be locally informed, and will instead effectively be a global variance measure. As $k$ increases, the $\mathcal{D}_i$ will instead be dominated by matches to nearby clusters, better capturing the smoothing needs in the immediate vicinity of $x_i$. Notably, the expansion is in the feature space alone, yielding theoretical results in the case of 1-mixup on generalization and adversarial robustness.

An alternative approach by Carratino et al. (2020) characterizes mixup as a combination of a reversion to mean followed by random perturbation. In supplement Section I, we generalize their result to $k$-mixup via a locally-informed mean and covariance.

## 5 Implementation and Experiments

### 5.1 $k$-mixup Implementation & Computational Cost

Figure 5 shows pseudocode for our implementation of one epoch of $k$-mixup training, in the style of Figure 1 of Zhang et al. (2018).[4] Also provided in Algorithm 1 is the procedure used in the experiments to adjust $\alpha$ as $k$ increases (denoted $\alpha_k$). While $\xi$ cannot be evaluated in closed form as a function of $\alpha$, since $\xi$ monotonically increases with $\alpha$ it is sufficient to iteratively increase $\alpha_k$ until the desired $\xi$ is reached. The while loop involves only computing the empirical expectation of a simple function of a scalar beta-distributed random variable and is therefore fast to compute.

---

[4]Python code for applying $k$-mixup to CIFAR10 can be found at `https://github.com/AnmingGu/kmixup-cifar10`.

```
# y1, y2 should be one-hot vectors
for (x1, y1), (x2, y2) in zip(loader1, loader2):
    idx = numpy.zeros_like(y1)
    for i in range(x1.shape[0] // k):
        cost = scipy.spatial.
            distance_matrix(x1[i * k:(i+1) * k], x2[i * k:(i+1) * k])
        _, ix = scipy.optimize.linear_sum_assignment(cost)
        idx[i * k:(i+1) * k] = ix + i * k
    x2 = x2[idx]
    y2 = y2[idx]
    lam = numpy.random.beta(alpha, alpha)
    x = Variable(lam * x1 + (1 - lam) * x2)
    y = Variable(lam * y1 + (1 - lam) * y2)
    optimizer.zero_grad()
    loss(net(x), y).backward()
    optimizer.step()
```

Figure 5: $k$-mixup implementation

---

**Algorithm 1** Choosing $\alpha_k$ to maintain constant $\xi$

---

**Require:** Chosen $\alpha_1$, desired $k$, trials $N$, constant $c > 1$, threshold $\gamma$, dataset of interest.
 1: Using $N$ trials, compute $\bar{\xi}_1$ as the empirical average of the squared distance between two random points in the dataset.
 2: Using $N/k$ trials, compute $\bar{\xi}_k$ as the empirical average of the squared Wasserstein-2 distance between two random $k$-samples drawn from the dataset.
 3: Using $N$ trials, compute empirical average $\bar{\lambda}_1$ of $\min(\lambda_1^2, (1 - \lambda_1)^2)$ where $\lambda_1 \sim \beta(\alpha_1, \alpha_1)$.
 4: $\xi \leftarrow \bar{\xi}_1 \bar{\lambda}_1$
 5: Initialize $\alpha_k = \alpha$, $\bar{\lambda}_k = \bar{\lambda}_1$.
 6: **while** $\bar{\lambda}_k \bar{\xi}_k < \xi$ and $\alpha_k < \gamma$ **do**
 7:     $\alpha_k \leftarrow \alpha_k c$
 8:     Using $N$ trials, compute empirical average $\bar{\lambda}_k$ of $\min(\lambda_k^2, (1 - \lambda_k)^2)$ where $\lambda_k \sim \beta(\alpha_k, \alpha_k)$.
 9: **end while**
10: Output $\alpha_k$.

---

**Computational cost.** While the cost of the Hungarian algorithm is $O(k^3)$, it provides $k$ data points for regularization, yielding an amortized $O(k^2)$ complexity per data point. For the smaller values of $k$ that we empirically consider, approximate Sinkhorn-based methods (Cuturi, 2013) are slower in practice, and the Hungarian cost remains small relative to that of gradient computation (e.g. for CIFAR-10 and $k = 32$, the Hungarian algorithm costs 0.69 seconds per epoch in total). Computing the distance matrix input to the OT matching costs $O(k^2 d)$ where $d$ is dimension, yielding an amortized $O(kd)$ complexity per data point. With the high dimensionality of CIFAR, a naive GPU implementation of this step of the computation adds about 0.5 seconds per epoch. Note that, on our hardware, the overall cost of an epoch is greater than 30 seconds. Moreover, training convergence speed is unaffected in terms of epoch count, unlike in manifold mixup (Verma et al., 2018), which in our experience converges slower and has larger computational overhead. Overall, therefore, there is little computational downside to generalizing 1-mixup to $k$-mixup, and, as we will see, the potential for performance gains.

## 5.2 Empirical Results

The simplicity of our method allows us to test the efficacy of $k$-mixup over a wide range of datasets and domains: 5 standard benchmark image datasets, 5 UCI tabular datasets, and a speech dataset; employing a variety of fully connected and convolutional neural network architectures. Across these experiments we find that $k$-mixup for $k > 1$ consistently improves upon 1-mixup after hyperparameter optimization. The gains

are generally on par with the improvements of 1-mixup over no-mixup (ERM). Interestingly, when the mean perturbation distances (squared) are constrained to be at a fixed level $\xi^2$, $k$-mixup for $k > 1$ still usually improves upon 1-mixup, with the improvement especially stark for larger perturbation sizes. In the tables below, $\xi$ is used to denote the root mean square of the perturbation distances, and $\alpha$ denotes the $\alpha$ used in the $k = 1$ case to achieve this $\xi$. We show here sweeps over $k$ and $\xi$, choosing $k$ and $\xi$ in practice is discussed in Supplement Section A.

Unless otherwise stated, our training is done over 200 epochs via a standard SGD optimizer, with learning rate 0.1 decreased at epochs 100 and 150, momentum 0.9, and weight decay $10^{-4}$. Note that throughout, we focus on comparing ERM, 1-mixup, and $k$-mixup on different architectures, rather than attempting to achieve a new state of the art on these benchmarks.

**Image datasets.** Our most extensive testing was done on image datasets, given their availability and the efficacy of neural networks in this application domain. In Table 1, we show our summarized error rates across various benchmark datasets and network architectures (all results are averaged over 20 random trials).[5] For each combination of dataset and architecture, we report the improvement of $k$-mixup over 1-mixup, allowing for optimization over relevant parameters. In this table, hyperparameter optimization refers to optimizing over a discrete set of the designated hyperparameter values listed in the detailed hyperparameter result tables shown below for each dataset and setting. As can be seen, the improvement is on par with that of 1-mixup over ERM. We also show the performance of $k = 16$ (a heuristic choice for $k$), which performed well across all experiments, outperforming 1-mixup in nearly all instances. This $k = 16$-mixup performance is notable as it only requires optimizing a single hyperparameter $\alpha$ / $\xi$, the same hyperparameter used for 1-mixup.

Table 1: Summary of image dataset error rate results (in percent, all results averaged over 20 trials). Note that for each setting, $k$-mixup with hyperparameter optimization outperforms both ERM and optimized 1-mixup. We also show 16-mixup as a heuristic that only involves optimizing $\alpha$ / $\xi$, i.e. the hyperparameter space is the same as 1-mixup. This heuristic outperforms ERM and 1-mixup in all but one instance, where it matches the performance of 1-mixup. Note that, on average, the improvement of $k$-mixup over 1-mixup tends to be similar in magnitude to the improvement of 1-mixup over ERM.

| Dataset (Confidence) | Architecture | ERM | 1-mixup ($\xi$ opt.) | $k$-Mixup ($k, \xi$ opt.) | 16-Mixup (heuristic, $\xi$ opt.) |
|---|---|---|---|---|---|
| MNIST ($\pm.02$) | LeNet | 0.95 | 0.76 | **0.66** | 0.74 |
| CIFAR-10 ($\pm.03$) | ResNet18 | 5.6 | 4.18 | **4.02** | **4.02** |
| ($\pm.03$) | DenseNet-BC-190 | 3.70 | 3.29 | **2.85** | **2.85** |
| ($\pm.09$) | WideResNet-101 | 11.6 | 11.53 | **11.25** | **11.25** |
| CIFAR-100 ($\pm.05$) | DenseNet-BC-190 | 18.91 | 18.91[6] | **18.31** | **18.31** |
| SVHN ($\pm.02$) | ResNet18 | 3.37 | 2.93 | **2.78** | 2.93 |
| Tiny ImageNet ($\pm.06$) | ResNet18 | 38.50 | 37.58 | **35.67** | **35.67** |

Full parameter sweeps for all settings shown in Table 1 are in Appendix C, with two presented below that are indicative of the general pattern that we see as we vary $\alpha$ / $\xi$ and $k$. In Table 2, we show results for MNIST (LeCun & Cortes, 2010) with a slightly modified LeNet architecture to accommodate grayscale images, and for Tiny ImageNet with a Pre-Act ResNet-18 architecture as in Zhang et al. (2018). Each table entry is averaged over 20 Monte Carlo trials. Here and throughout the remainder of the paper, error bars are reported for the performance results.[7] In the MNIST results, for each $\xi$ the best generalization performance is for some $k > 1$, i.e.

Table 2: Results for MNIST with a LeNet architecture (no mixup (ERM) error: 0.95%), averaged over 20 trials ($\pm.02$ confidence on test error). Note the best $k$-mixup beats the best 1-mixup by 0.1%; on the same order as the 0.19% improvement of 1-mixup over ERM.

| | $\alpha = .05$ | $\alpha = .1$ | $\alpha = .2$ | $\alpha = .5$ | $\alpha = 1$ | $\alpha = 10$ | $\alpha = 100$ |
|---|---|---|---|---|---|---|---|
| $k$ | $\xi = 0.5$ | $\xi = 0.6$ | $\xi = 0.8$ | $\xi = 1.2$ | $\xi = 1.4$ | $\xi = 2.1$ | $\xi = 2.2$ |
| 1 | 0.79 | 0.79 | 0.76 | 0.86 | 0.83 | 1.05 | 1.26 |
| 2 | 0.85 | 0.79 | 0.90 | 0.76 | 0.83 | 0.94 | 1.14 |
| 4 | 0.91 | 0.80 | 0.83 | 0.81 | 0.85 | 0.89 | 0.95 |
| 8 | 0.80 | 0.81 | 0.78 | 0.79 | 0.78 | 0.81 | 0.83 |
| 16 | 0.77 | 0.82 | 0.80 | 0.75 | 0.80 | 0.74 | 0.75 |
| 32 | 0.79 | 0.75 | 0.77 | 0.79 | 0.80 | 0.76 | 0.71 |
| 64 | 0.80 | 0.82 | 0.77 | 0.78 | 0.78 | **0.66** | 0.71 |

---

[5]As a result, Table 1 summarizes results from training 3680 neural network models in total.

[6]Achieved with $\alpha = 0$, i.e. equivalent to ERM.

[7]These error bars are the one standard deviation of the Monte Carlo average, where for brevity we report only the worst such variation over the elements of the corresponding results table.

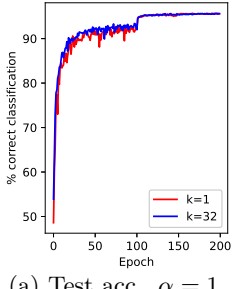
(a) Test acc., $\alpha = 1$.

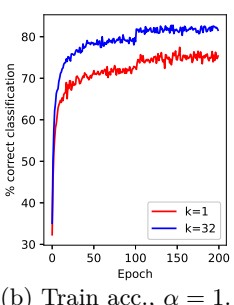
(b) Train acc., $\alpha = 1$.

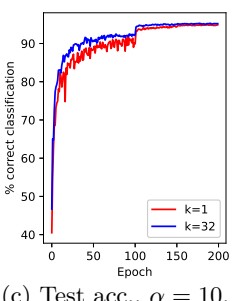
(c) Test acc., $\alpha = 10$.

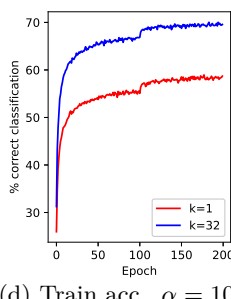
(d) Train acc., $\alpha = 10$.

Figure 6: Training convergence of $k = 1$ and $k = 32$-mixup on CIFAR-10, averaged over 20 random trials. Both train at roughly the same rate ($k = 32$ slightly faster), the train accuracy discrepancy is due to the more class-accurate vicinal training distribution created by higher $k$-mixup.

$k$-mixup outperforms 1-mixup for all perturbation sizes. The lowest error rate overall is achieved with $k = 64$ and $\xi = 2.1$: 0.66%, an improvement of 0.1% over the best 1-mixup and an improvement of 0.29% over ERM.

For Tiny ImageNet, we again see that for each $\alpha$ / $\xi$, the best generalization performance is for some $k > 1$ (note that the $\xi$ are larger for Tiny ImageNet than MNIST due to differences in normalization and size of the images). The lowest error rate overall is achieved with $k = 16$ and $\xi = 1000$: 35.67%, and improvement of 1.91% over the best 1-mixup and an improvement of 2.83% over ERM. In both datasets, we see that for each perturbation size value $\xi$, the best generalization performance is for some $k > 1$, but that the positive effect of increasing $k$ is seen most clearly for larger $\xi$. This strongly supports the notion that $k$-mixup is significantly more effective than 1-mixup at designing an effective vicinal distribution for larger perturbation budgets. The lowest overall error rates are achieved for relatively high $\xi$ and high $k$.

Table 3: Results for Tiny ImageNet with a ResNet18 architecture (no mixup (ERM) error: 38.50%), averaged over 20 trials ($\pm.06$ confidence on test error). Note the best $k$-mixup beats the best 1-mixup by 1.91%; better than the 0.92% improvement of 1-mixup over ERM.

| $k$ | $\alpha = .1$ $\xi = 9.7$ | $\alpha = .2$ $\xi = 13$ | $\alpha = .5$ $\xi = 18$ | $\alpha = 1$ $\xi = 22$ | $\alpha = 10$ $\xi = 32$ |
|---|---|---|---|---|---|
| 1 | 38.47 | 38.26 | 37.79 | 37.61 | 37.58 |
| 2 | 38.48 | 38.06 | 37.89 | 37.43 | 37.00 |
| 4 | 38.51 | 38.13 | 37.67 | 37.44 | 36.21 |
| 8 | 38.42 | 38.05 | 37.67 | 37.19 | 35.86 |
| 16 | 38.47 | 38.10 | 37.60 | 37.28 | **35.67** |
| 32 | 38.45 | 38.04 | 37.60 | 37.29 | 35.88 |

Additionally, we show training curves (performance as a function of epoch) demonstrating that the use of $k$-mixup does not affect training convergence. These are shown for ResNet-18 on CIFAR-10 in Figure 6, which is indicative of what is seen across all our experiments. The training speeds (test accuracy in terms of epoch) for 1-mixup and 32-mixup show no loss of convergence speed with $k = 32$-mixup, with (if anything) $k = 32$ showing a slight edge. The discontinuity at epoch 100 is due to our reduction of the learning rate at epochs 100 and 150 to aid convergence (used throughout our image experiments). The train accuracy shows a similar convergence profile between 1- and 32-mixup; the difference in absolute accuracy here (and the reason it is less than the test accuracy) is because the training distribution is the mixup-modified vicinal distribution. The curve for $k = 32$ is higher, especially for $\alpha = 10$, because the induced vicinal distribution and labels are more consistent with the true distribution, due to the better matches from optimal transport. The large improvement in train accuracy is remarkable given the high dimension of the CIFAR-10 data space, since it indicates that $k = 32$-mixup is able to find significantly more consistent matches than $k = 1$-mixup.

**UCI datasets.** To demonstrate efficacy on non-image data, we tried $k$-mixup on UCI datasets (Dua & Graff, 2017) of varying size and dimension: Iris (150 instances, dim. 4), Breast Cancer Wisconsin-Diagnostic (569 instances, dim. 30), Abalone (4177 instances, dim. 8), Arrhythmia (452 instances, dim. 279), and Phishing (11055 instances, dim. 30). For Iris, we used a 3-layer network with 120 and 84 hidden units; for Breast Cancer, Abalone, and Phishing, we used a 4-layer network with 120, 120, and 84 hidden units; and lastly, for Arrhythmia we used a 5-layer network with 120, 120, 36, and 84 hidden units. For these datasets we used a learning rate of 0.005 instead of 0.1. Each entry is averaged over 20 Monte Carlo trials. Test error rate is shown in Figure 4. $k$-mixup improves over 1-mixup in each case, although for Arrhythmia no mixup

outperforms both. In these small datasets, the best performance seems to be achieved with relatively small $\alpha$ $(0.05, 0.1)$ and larger $k$ (4 or greater).

Table 4: Test error on UCI datasets using fully connected networks, averaged over 20 random trials.

| Dataset | $\alpha$ | $\xi$ | $k=1$ | $k=2$ | $k=4$ | $k=8$ | $k=16$ |
|---|---|---|---|---|---|---|---|
| Abalone | 0.05 | .07 | 71.37 | 71.32 | 71.23 | 71.54 | 71.47 |
| (ERM: 72.32) | 0.1 | .09 | 71.32 | 71.41 | 71.82 | 71.05 | 71.07 |
| | 1.0 | .20 | 71.56 | 71.85 | 71.26 | 71.29 | **70.90** |
| Arrhythmia | 0.05 | 13.97 | 35.15 | 34.84 | 33.88 | 35.16 | 34.97 |
| (ERM: **32.06**) | 0.1 | 17.83 | 34.14 | 36.13 | 34.29 | 35.64 | **33.39** |
| | 1.0 | 42.28 | 34.25 | 35.51 | 33.78 | 33.55 | 33.64 |
| Cancer | 0.05 | 19 | 10.32 | 9.08 | 9.52 | **8.63** | 9.35 |
| (ERM: 11.25) | 0.1 | .26 | 10.42 | 9.75 | 9.92 | 9.14 | 10.25 |
| | 1.0 | .59 | 12.31 | 10.83 | 10.23 | 9.85 | 9.15 |

| Dataset | $\alpha$ | $\xi$ | $k=1$ | $k=2$ | $k=4$ | $k=8$ | $k=16$ |
|---|---|---|---|---|---|---|---|
| Iris | 0.05 | .07 | 6.70 | 4.20 | 3.30 | **2.50** | 3.70 |
| (ERM: 4.10) | 0.1 | .10 | 6.00 | 2.90 | 2.70 | 3.50 | 3.60 |
| | 1.0 | .22 | 7.00 | 3.90 | 2.90 | 3.80 | 2.90 |
| Phishing | 0.05 | .28 | 3.30 | 3.14 | **3.05** | 3.17 | 3.06 |
| (ERM: 3.43) | 0.1 | .38 | 3.30 | 3.36 | 3.35 | 3.34 | 3.26 |
| | 1.0 | .85 | 4.69 | 4.55 | 4.27 | 4.18 | 4.20 |

**Speech dataset**. Performance is also tested on a speech dataset: Google Speech Commands (Warden, 2018). Results with a simple LeNet architecture are in Table 5. Each table entry is averaged over 20 Monte Carlo trials ($\pm$.014 confidence on test performance). We augmented the data in the same way as Zhang et al. (2018), i.e. we sample the spectrograms from the data using a sampling rate of 16 kHz and equalize their sizes at $160 \times 101$. We also use similar training parameters: we train for 30 epochs using a learning rate of $3 \times 10^{-3}$ that is divided by 10 every 10 epochs. The improvement of the best $k$-mixup over the best 1-mixup is 1.11%, with the best $k$-mixup performance being for $k = 16$ and large $\alpha$ / $\xi$. Note this improvement is greater than the 0.83% improvement of 1-mixup over ERM.

**Toy datasets.** Note that for completeness, in Appendix B we provide quantitative results for the toy datasets of Figures 1, 2, and 3, confirming the qualitative analysis therein.

**Distribution shift.** As mentioned above, a key benefit of mixup is that it (approximately) smoothly linearly interpolates between classes and thereby should provide a degree of robustness to distribution shifts that involve small perturbation of the features Zhang et al. (2018). Here we test two such forms of distribution shift: additive Gaussian noise and FGSM white box adversarial attacks (a full exploration of the infinite set of possible distribution shifts is beyond the scope of this paper). Additive Gaussian noise can arise from unexpected noise in the imaging sensor, and testing adversarial FGSM attacks on mixup was introduced in the original mixup paper Zhang et al. (2018) as a much more difficult perturbation test than i.i.d. additive noise. As in Zhang et al. (2018), we limit the experiment to

Table 5: Google Speech Commands test error using LeNet architecture (no mixup (ERM) error: 12.26%), averaged over 20 Monte Carlo trials. Note the best $k$-mixup beats the best 1-mixup by 1.11%, greater than the 0.83% improvement of 1-mixup over ERM.

| $k$ | $\alpha = .1$ $\xi = 14$ | $\alpha = .2$ $\xi = 19$ | $\alpha = .5$ $\xi = 26$ | $\alpha = 1$ $\xi = 31$ | $\alpha = 10$ $\xi = 45$ |
|---|---|---|---|---|---|
| 1 | 11.89 | 11.55 | 11.43 | 11.55 | 12.16 |
| 2 | 11.76 | 11.48 | 11.32 | 11.38 | 11.62 |
| 4 | 11.68 | 11.36 | 11.18 | 11.19 | 11.20 |
| 8 | 11.83 | 11.36 | 11.14 | 11.14 | 10.84 |
| 16 | 11.72 | 11.36 | 11.11 | 11.04 | **10.32** |

basic FGSM attacks, since iterative PGD attacks are too strong, making any performance improvements seem less relevant in practice and calling into question the realism of using PGD as a proxy for distribution shifts.

Additive Gaussian noise results for CIFAR-10 and various levels $\epsilon$ of noise and mixup parameter $\alpha$ are shown in Figure 6.[8] Note that the $k = 16$ outperforms $k = 1$ for all noise levels by as much as 4.29%, and ERM by as much as 16%.

Following the experimental setup of Zhang et al. (2018), Figure 7 shows results on white-box adversarial attacks generated by the FGSM method (implementation of Kim (2020)[9]) for various values of maximum adversarial perturbation. As in Zhang et al. (2018), the goal of this test is not to achieve results comparable to the much more computationally intensive methods of adversarial training. This would be impossible for a non-adversarial regularization approach. We show CIFAR-10 accuracy on white-box FGSM adversarial data (10000 points), where the maximum adversarial perturbation is set to $\epsilon/255$; performance is averaged over 30 Monte Carlo trials ($\pm$0.6 confidence). Note that the highest $k$ outperforms $k = 1$ uniformly by as much

---

[8]Smaller values of $\alpha$ were tried, but this decreased performance for all $k$ values.

[9]Software has MIT License

Table 6: Additive noise: Error for CIFAR-10 ResNet-18 with additive Gaussian noise of standard deviation $\epsilon/255$ (results averaged over 30 trials, $\pm 0.6$ confidence). $k = 16$-mixup outperforms 1-mixup and ERM in each instance (best performance for each $\epsilon$ in bold).

| | | $\alpha = 2$ | | | $\alpha = 5$ | | | $\alpha = 10$ | | |
|---|---|---|---|---|---|---|---|---|---|---|
| Noise | ERM | $k=1$ | $k=4$ | $k=16$ | $k=1$ | $k=4$ | $k=16$ | $k=1$ | $k=4$ | $k=16$ |
| $\epsilon=8$ | 15.72 | 11.10 | 11.65 | **11.00** | 11.39 | 11.19 | 11.28 | 11.88 | 11.28 | 11.53 |
| $\epsilon=10$ | 22.21 | 15.47 | 16.06 | 14.34 | 15.78 | 14.69 | **14.20** | 15.43 | 14.56 | 14.37 |
| $\epsilon=12$ | 29.55 | 21.69 | 21.75 | 18.75 | 21.19 | 19.06 | 17.71 | 20.56 | 18.45 | **17.69** |
| $\epsilon=14$ | 37.93 | 29.69 | 28.34 | 23.86 | 27.43 | 23.72 | 21.62 | 25.90 | 22.62 | **21.61** |

as 6.12%, and all $k > 1$-mixups outperform or statistically match $k = 1$-mixup for all attack sizes. Similar results for MNIST are in Table 7(b), with the FGSM attacks being somewhat less effective. Here again, the highest $k$ performs the best for all attack sizes.

The improved robustness shown by $k$-mixup speaks to a key goal of mixup, that of smoothing the predictions in the parts of the data space where no/few labels are available. This smoothness should make adversarial attacks require greater magnitude to successfully "break" the model.

Table 7: Adversarial shifts: error on white-box FGSM attacks. Parameter $\alpha$ chosen to maximize $k = 1$ performance. Large $k$-mixup outperforms 1-mixup in each instance.

| $k$ | $\epsilon=.5$ | $\epsilon=1$ | $\epsilon=2$ | $\epsilon=4$ | $\epsilon=8$ | $\epsilon=16$ |
|---|---|---|---|---|---|---|
| 1 | 20.63 | 26.25 | 31.07 | 36.72 | 48.55 | 74.78 |
| 2 | 20.40 | 25.70 | 29.89 | 34.34 | 43.78 | 72.22 |
| 4 | 20.73 | 26.29 | 30.59 | 34.94 | 43.94 | 71.20 |
| 8 | 20.58 | 26.06 | 30.31 | 34.50 | 43.51 | 70.18 |
| 16 | **20.21** | **25.50** | **29.57** | **33.80** | **43.32** | **68.66** |

| $k$ | $\epsilon=.5$ | $\epsilon=1$ | $\epsilon=2$ | $\epsilon=4$ | $\epsilon=8$ | $\epsilon=16$ |
|---|---|---|---|---|---|---|
| 1 | 1.23 | 1.66 | 2.37 | 3.88 | 7.70 | 18.10 |
| 2 | 1.12 | 1.50 | 2.11 | 3.58 | 7.44 | 19.81 |
| 4 | 1.07 | 1.37 | 1.94 | 3.25 | 7.30 | 22.02 |
| 8 | 1.08 | 1.40 | 1.88 | 3.12 | 6.92 | 21.00 |
| 16 | 0.88 | 1.11 | 1.50 | 2.42 | 5.28 | 16.44 |
| 32 | **0.85** | **1.07** | **1.39** | **2.27** | **4.91** | **14.95** |

(a) CIFAR-10 ($\pm 0.6$ confidence). $\alpha = 0.5$.      (b) MNIST ($\pm 0.1$ confidence). $\alpha = 10$.

**Manifold mixup.** We have also compared to manifold mixup (Verma et al., 2018), which aims to interpolate data points in deeper layers of the network to get more meaningful interpolations. This leads to the natural idea of doing $k$-mixup in these deeper layers. We use the settings in Verma et al. (2018), i.e., for ResNet18, the mixup layer is randomized (coin flip) between the input space and the output of the first residual block.[10]

Results for CIFAR-10 with a ResNet18 architecture are in Table 8. Note that in this experiment, the $\alpha$'s shown are used for all $k$ without adjustment since the matches across $k$-samples happen at random layers and cannot be standardized as simply. Numbers in this experiment are averaged over 20 Monte Carlo trials ($\pm .03$ confidence on test performance). Manifold 1-mixup is matched by manifold $k$-mixup with $k = 4$, but both are outperformed by standard $k$-mixup (Table 1). **We therefore do not find a benefit to using "$k$-manifold-mixup" over our proposed $k$-mixup.** We also tried randomizing over (a) the outputs of all residual blocks and (b) the outputs of (lower-dimensional) deep residual blocks only, but found that performance of both manifold 1-mixup and manifold $k$-mixup degrades in these cases. This latter observation underscores that mixup in hidden layer manifolds is not guaranteed to be effective and can require tuning.

Table 8: Comparison to manifold mixup approach. "$k$-manifold mixup" test error on CIFAR-10 using ResNet18 architecture, averaged over 20 Monte Carlo trials ($\pm .03$ confidence). Compare Table 1 and observe that this "manifold $k$-mixup" matches the standard manifold mixup but **both underperform our proposed $k$-mixup, which achieves a 4.02% error rate with $k = 16$.**

| $k$ | $\alpha = .1$ | $\alpha = .2$ | $\alpha = .5$ | $\alpha = 1$ | $\alpha = 10$ |
|---|---|---|---|---|---|
| 1 | 5.00 | 4.73 | 4.41 | 4.26 | 4.81 |
| 2 | 5.09 | 4.78 | 4.44 | 4.31 | 4.71 |
| 4 | 5.01 | 4.78 | 4.45 | **4.25** | 5.02 |
| 8 | 5.08 | 4.78 | 4.42 | 4.28 | 4.71 |
| 16 | 5.14 | 4.82 | 4.42 | 4.31 | 4.53 |
| 32 | 5.14 | 4.91 | 4.58 | 4.47 | 4.54 |

---

[10]See `https://github.com/vikasverma1077/manifold_mixup`

# 6 Conclusions and Future Work

The experiments above demonstrate that $k$-mixup improves the generalization and robustness gains achieved by 1-mixup. This is seen across a diverse range of datasets and network architectures. It is simple to implement, adds little computational overhead to conventional 1-mixup training, and may also be combined with related mixup variants. As seen in the theory presented in Sections 3 and 4, as $k$ increases, the induced regularization more accurately reflects the local structure of the training data, especially in the manifold support and clustered settings. Empirical results show that performance is relatively robust to variations in $k$, especially when normalizing for similar perturbation distances (squared), ensuring that extensive tuning is not necessary.

With the notable exception of the larger improvement on Tiny ImageNet, our experiments show the improvement on high-dimensional datasets is sometimes smaller than on lower dimensional datasets (recall that classic mixup also has somewhat small gains over ERM in these settings). This difference may be influenced by the diminishing value of Euclidean distance for characterizing dataset geometry in high dimensions (Aggarwal et al., 2001), but intriguingly this effect was not remedied by doing OT in the lower-dimensional manifolds created by the higher layers in our manifold mixup experiments. In future work we will consider alternative metric learning strategies, with the goal of identifying alternative high-dimensional metrics for displacement interpolation of data points.

## Acknowledgements

The MIT Geometric Data Processing group acknowledges the generous support of Army Research Office grants W911NF2010168 and W911NF2110293, of Air Force Office of Scientific Research award FA9550-19-1-031, of National Science Foundation grants IIS-1838071 and CHS-1955697, from the CSAIL Systems that Learn program, from the MIT–IBM Watson AI Laboratory, from the Toyota–CSAIL Joint Research Center, from a gift from Adobe Systems, and from a Google Research Scholar award.

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

Table 9: Test error on toy datasets, averaged over 5 Monte Carlo trials.

| $k$ | $\alpha=.25$ $\xi=.10$ | $\alpha=1$ $\xi=.15$ | $\alpha=4$ $\xi=.20$ | $\alpha=16$ $\xi=.23$ | $\alpha=64$ $\xi=.25$ | $k$ | $\alpha=.25$ $\xi=.50$ | $\alpha=1$ $\xi=.78$ | $\alpha=4$ $\xi=1.0$ | $\alpha=16$ $\xi=1.1$ | $\alpha=64$ $\xi=1.2$ | $k$ | $\alpha=.25$ $\xi=1.2$ | $\alpha=1$ $\xi=1.9$ | $\alpha=4$ $\xi=2.4$ | $\alpha=16$ $\xi=2.8$ | $\alpha=64$ $\xi=3.0$ |
|---|---|---|---|---|---|---|---|---|---|---|---|---|---|---|---|---|---|
| 1 | 5.01 | 7.88 | 9.85 | 10.28 | 9.75 | 1 | 0.00 | 0.00 | 6.79 | 49.60 | 49.80 | 1 | 0.83 | 9.52 | 31.84 | 37.38 | 36.18 |
| 2 | 4.28 | 5.19 | 5.87 | 6.19 | 5.79 | 2 | 0.01 | 0.00 | 1.53 | 47.82 | 49.74 | 2 | 0.49 | 3.30 | 28.41 | 32.21 | 32.75 |
| 4 | 2.93 | 3.34 | 4.10 | 3.86 | 3.92 | 4 | 0.00 | 0.00 | 1.41 | 8.25 | 9.58 | 4 | **0.32** | 1.34 | 15.49 | 27.53 | 29.23 |
| 8 | 2.50 | 2.77 | 3.24 | 3.15 | 2.78 | 8 | 0.00 | 0.00 | 0.38 | 0.62 | 0.63 | 8 | 0.35 | 0.45 | 2.67 | 8.14 | 2.76 |
| 16 | **2.42** | **2.26** | **2.54** | **2.43** | **2.30** | 16 | 0.00 | 0.00 | **0.08** | **0.07** | **0.22** | 16 | 0.40 | **0.33** | **0.45** | **0.27** | **0.27** |
| | (a) One Ring | | | | | | (b) Four Bars | | | | | | (c) Swiss Roll | | | | |

Pete Warden. Speech commands: A dataset for limited-vocabulary speech recognition. *arXiv preprint arXiv:1804.03209*, 2018.

Jonathan Weed and Francis Bach. Sharp asymptotic and finite-sample rates of convergence of empirical measures in wasserstein distance. *Bernoulli*, 25(4A):2620–2648, 2019.

Sangdoo Yun, Dongyoon Han, Seong Joon Oh, Sanghyuk Chun, Junsuk Choe, and Youngjoon Yoo. CutMix: Regularization Strategy to Train Strong Classifiers With Localizable Features. pp. 6023–6032, 2019. URL https://openaccess.thecvf.com/content_ICCV_2019/html/Yun_CutMix_Regularization_Strategy_to_Train_Strong_Classifiers_With_Localizable_Features_ICCV_2019_paper.html.

Hongyi Zhang, Moustapha Cisse, Yann N. Dauphin, and David Lopez-Paz. mixup: Beyond Empirical Risk Minimization. February 2018. URL https://openreview.net/forum?id=r1Ddp1-Rb.

Linjun Zhang, Zhun Deng, Kenji Kawaguchi, Amirata Ghorbani, and James Zou. How Does Mixup Help With Robustness and Generalization? September 2020. URL https://openreview.net/forum?id=8yKEo06dKNo.

# A    Hyperparameter tuning

While as noted in the main text, the large value of $k = 16$ with $\alpha$ optimized consistently performs well, increasing $k$ does not always improve performance monotonically. This, however, is to be expected in any real data scenario. Hence in practice, it is often appropriate to search over $k$. This is not too difficult as in our experiments we found it sufficient to only try powers of 2, and performance generally is smoothly varying over $\alpha$. Several search approaches work well:

1. **Very simple**: set $\alpha = 1$ and search over $k$ (powers of 2). It can be seen from our experiments that except for one UCI dataset this approach outperforms or matches the 1-mixup performance at any $\alpha$.

2. **More complex but still relatively simple**: search over $\xi$ for 1-mixup, fix the best such $\xi$, and then search over $k$ (in powers of 2 up to $k = 16$ or 32). This approach always outperforms 1-mixup and does not add much hyperparameter search overhead.

3. **Full hyperparameter grid search**: Not too expensive for many neural networks, for instance a full grid search (with $k$ powers of 2) for ResNet-18 on CIFAR-10 requires only a few hours of time on our medium-sized cluster. If the model will be deployed in high-stakes or high-volume applications, full search would be feasible even for larger networks.

# B    Test error on toy datasets.

For completeness, we provide quantitative results for the toy datasets of Figures 1, 2, and 3 (denoted "One Ring," "Four Bars," and "Swiss Roll") in Table 9, continuing the intuition-building discussion in each of those figures. We used a fully-connected 3-layer neural network (130 and 120 hidden units). As the datasets are well-clustered and have no noise, smoothing is not needed for generalization and performance without mixup is typically 100%. Rather than beating ERM, applying mixup to these datasets instead aims to build

intuition, providing a view to the propensity of each variant to oversmooth, damaging performance. For each dataset and each perturbation size $\xi$, higher $k$-mixup outperforms 1-mixup, with $k = 16$ providing the best performance in all but one instance. These results quantitatively confirm the intuition built in Figures 1, 2, and 3 that $k$-mixup regularization more effectively preserves these structures in data, limiting losses from oversmoothing at a fixed mean perturbation size.

## C  Additional Image Dataset Parameter Sweeps

In this section, we show the parameter sweep tables for the remaining rows of Table 1.

Table 10 below shows results for CIFAR-10, using the PreAct ResNet-18 architecture as in Zhang et al. (2018). Again, increasing $k$ for fixed $\xi$ tends to improve generalization performance, especially in the high-$\xi$ regime. The best performance overall is achieved at $k = 16$ with $\xi = 16.7$. While the best $k$-mixup performance exceeds that of the best 1-mixup by 0.16%, recall that in this setting, 1-mixup outperforms ERM by only 1.4% (Zhang et al., 2018), so when combined with the low overall error rate, small gains are not surprising. Results for DenseNet and WideResNet architectures can be found in Table 11, with the best $k$-mixup outperforming the best 1-mixup by 0.44% and 0.28% respectively. Note that in the case of DenseNet, $k = 16$ outperforms or (statistically) matches $k = 1$ for all values of $\xi$.

Table 10: Results for CIFAR-10 with ResNet18 architecture (no mixup (ERM) error: 5.6%), averaged over 20 Monte Carlo trials ($\pm.03$ confidence on test performance). Difference between best $k$-mixup and best 1-mixup is 0.16%; for fixed high $\alpha$ ($\alpha = 100$), the improvement increases to 1.19%.

|   | $\alpha = .05$ | $\alpha = .1$ | $\alpha = .2$ | $\alpha = .5$ | $\alpha = 1$ | $\alpha = 10$ | $\alpha = 100$ |
|---|---|---|---|---|---|---|---|
| $k$ | $\xi = 5.6$ | $\xi = 7.4$ | $\xi = 10.0$ | $\xi = 13.8$ | $\xi = 16.7$ | $\xi = 24.0$ | $\xi = 27.2$ |
| 1 | 5.01 | 4.68 | 4.41 | 4.24 | 4.18 | 4.95 | 5.67 |
| 2 | 4.92 | 4.69 | 4.46 | 4.13 | 4.03 | 4.58 | 5.46 |
| 4 | 4.88 | 4.68 | 4.52 | 4.13 | 4.03 | 4.48 | 5.19 |
| 8 | 4.91 | 4.77 | 4.51 | 4.21 | 4.08 | 4.42 | 4.92 |
| 16 | 4.92 | 4.77 | 4.48 | 4.23 | **4.02** | 4.40 | 4.75 |
| 32 | 4.98 | 4.78 | 4.58 | 4.24 | 4.16 | 4.36 | 4.48 |

Table 11: CIFAR-10 test error for DenseNet-BC-190 and WideResNet-101 architectures. For DenseNet, the difference between best $k$-mixup and best 1-mixup is 0.44%, for fixed high $\xi$ ($\alpha = 20.2$), the improvement increases to 0.65%. For WideResNet, the difference between best $k$-mixup and best 1-mixup is 0.28%, for fixed high $\xi$ ($\xi = 20.2$), the improvement increases to 1.25%.

|   | $\alpha = .5$ | $\alpha = 1$ | $\alpha = 2$ | $\alpha = 4$ |
|---|---|---|---|---|
| $k$ | $\xi = 13.8$ | $\xi = 16.7$ | $\xi = 19.0$ | $\xi = 20.2$ |
| 1 | 3.35 | 3.29 | 3.42 | 3.57 |
| 16 | 3.38 | 2.98 | **2.85** | 2.92 |

(a) DenseNet-BC-190 architecture($\pm.03$ confidence, no mixup (ERM) error: 3.7%)

|   | $\alpha = .05$ | $\alpha = .2$ | $\alpha = .5$ | $\alpha = 1$ | $\alpha = 2$ | $\alpha = 4$ |
|---|---|---|---|---|---|---|
| $k$ | $\xi = 5.6$ | $\xi = 10.0$ | $\xi = 13.8$ | $\xi = 16.7$ | $\xi = 19.0$ | $\xi = 20.2$ |
| 1 | 11.54 | 11.53 | 11.62 | 11.78 | 12.25 | 12.99 |
| 4 | 11.36 | 11.47 | 11.27 | 11.41 | 11.59 | 12.16 |
| 16 | 11.53 | 11.59 | **11.25** | 11.34 | 11.38 | 11.74 |

(b) WideResNet-101 architecture ($\pm.09$ confidence, no mixup (ERM) error: 11.6%)

Table 12 shows results for CIFAR-100 and SVHN, with DenseNet-BC-190 and ResNet-18 architectures respectively. As before, for fixed $\xi$, the best performance is achieved for some $k > 1$. The improvement of the best $k$-mixup over the best 1-mixup is 2.14% for CIFAR-100 and 0.15% for SVHN. For fixed high $\alpha$, the $k$-mixup improvement over 1-mixup rises to 2.85% for CIFAR-100 and 0.52% for SVHN, possibly indicating that the OT matches yield better interpolation between classes, aiding generalization.

Table 12: CIFAR-100 (DenseNet-BC-190) and SVHN (ResNet18), averaged over 20 trials.

|  | $\alpha = .5$ | $\alpha = 1$ | $\alpha = 2$ | $\alpha = 4$ |
|---|---|---|---|---|
| $k$ | $\xi = 3.4$ | $\xi = 4.2$ | $\xi = 4.8$ | $\xi = 5.4$ |
| 1 | 28.52 | 27.76 | 20.45 | 21.53 |
| 8 | 18.35 | 18.78 | 19.16 | 19.71 |
| 16 | 18.33 | **18.31** | 18.53 | 18.85 |

(a) CIFAR-100 error ($\pm.05$ confidence, no mixup (ERM) error: 18.91%)

|  | $\alpha = .1$ | $\alpha = .2$ | $\alpha = .5$ | $\alpha = 1$ | $\alpha = 10$ |
|---|---|---|---|---|---|
| $k$ | $\xi = 3.8$ | $\xi = 5.1$ | $\xi = 6.9$ | $\xi = 9.5$ | $\xi = 11.6$ |
| 1 | 3.29 | 3.22 | 2.99 | 2.93 | 3.89 |
| 2 | 3.32 | 3.19 | 2.96 | 2.79 | 3.65 |
| 4 | 3.30 | 3.25 | 2.94 | **2.78** | 3.55 |
| 8 | 3.28 | 3.20 | 3.01 | 2.86 | 3.44 |
| 16 | 3.26 | 3.24 | 3.04 | 2.93 | 3.37 |

(b) SVHN error ($\pm.02$ confidence, no mixup (ERM) error: 3.37%)

# D  Proof of Proposition 3.2

Finite-sample convergence results for empirical measures (Theorem 9.1 of Solomon et al. (2020), Weed & Bach (2019)) imply that for an arbitrary sampling of $k$ points $\hat{\mu}_k$, we have

$$W_2^2(\mu, \hat{\mu}_k) \leq O(k^{-2/d})$$

with $1 - 1/k^2$ probability. The triangle inequality then implies that the Wasserstein-2 distance between our batches of $k$ samples will tend to 0 at the same asymptotic rate, specifically

$$W_2(\hat{\mu}_k^\gamma, \hat{\mu}_k^\zeta) \leq W_2(\mu, \hat{\mu}_k^\zeta) + W_2(\hat{\mu}_k^\gamma, \mu) \leq O(k^{-1/d})$$

again with $1 - 1/k^2$ probability.

Now, recalling the definition of the optimal coupling permutation $\sigma(i)$, we have

$$W_2^2(\hat{\mu}_k^\gamma, \hat{\mu}_k^\zeta) = \frac{1}{k} \sum_{i=1}^{k} \|x_i^\gamma - x_{\sigma(i)}^\zeta\|_2^2.$$

Hence,

$$\frac{1}{k} \sum_{i=1}^{k} \|x_i^\gamma - x_{\sigma(i)}^\zeta\|_2^2 \leq O(k^{-2/d})$$

with $1 - 1/k^2$ probability. Thus, for any $\mathcal{I} \subseteq [1, k]$ with $\|x_i^\gamma - x_{\sigma(i)}^\zeta\|_2^2 > k^{-1/d}$ for all $i \in \mathcal{I}$,

$$O(k^{-2/d}) \geq \frac{1}{k} \sum_{i=1}^{k} \|x_i^\gamma - x_{\sigma(i)}^\zeta\|_2^2 > \frac{|\mathcal{I}|}{k} k^{-1/d},$$

implying

$$|\mathcal{I}| < O(k^{1-1/d}) = k \cdot O(k^{-1/d}) \leq k\delta,$$

where the last inequality holds for any $\delta \in (0, 1]$ given $k$ large enough.

In essence, the fraction of matches that are long-distance (i.e. those in $\mathcal{I}$) is bounded by $\delta$ for large enough $k$. Under these conditions, the set of short-distance matches (i.e. those in $\bar{\mathcal{I}}$ the complement of $\mathcal{I}$) satisfy

$$|\bar{\mathcal{I}}| \geq (1 - \delta)k,$$

where by definition, $\|x_i^\gamma - x_{\sigma(i)}^\zeta\|_2^2 \leq k^{-1/d}$ for all $i \in \bar{\mathcal{I}}$. Crucially, for any chosen $\epsilon$, we have $k^{-1/d} < \epsilon$ for $k$ large enough, so for $k$ large enough

$$\|x_i^\gamma - x_{\sigma(i)}^\zeta\|_2 < \epsilon, \quad \forall i \in \bar{\mathcal{I}}.$$

By definition of $k$-mixup, for all $i \in \bar{\mathcal{I}}$, the corresponding mixup interpolated point will be an interpolation between $x_i^\gamma$ and $x_{\sigma(i)}^\zeta$, i.e.

$$\lambda x_i^\gamma + (1 - \lambda)x_{\sigma(i)}^\zeta$$

for $\lambda \in [0, 1]$. Since all $x_i^\gamma$ lie in $\mathcal{S}$ and for all $i \in \bar{\mathcal{I}}$, $\|x_i^\gamma - x_{\sigma(i)}^\zeta\|_2 < \epsilon$, the mixup interpolated point will lie in $B_\epsilon(\mathcal{S})$. The proposition results.

# E   Proof of Lemma 3.3

Firstly, observe that the maximum number of within-cluster matches is

$$\sum_i \min(r_i, s_i)$$

by definition, and the total number of matches overall must equal $\sum_i s_i = \sum_i r_i$. Hence the number of cross-cluster matchings must be larger than or equal to

$$\sum_i r_i - \sum_i \min(r_i, s_i) = \sum_i \max(0, r_i - s_i),$$

equivalently,

$$\sum_i s_i - \sum_i \min(r_i, s_i) = \sum_i \max(s_i - r_i, 0).$$

Averaging these implies the number of cross-cluster matchings cannot be smaller than

$$\frac{1}{2} \sum_i \left( \max(r_i - s_i, 0) + \max(0, s_i - r_i) \right) = \frac{1}{2} \sum_i |r_i - s_i|.$$

It remains to show that the number of cross-cluster matchings cannot exceed $\frac{1}{2} \sum_i |r_i - s_i|$. We argue by contradiction and prove the result for $m = 2$ first. Suppose that the number of cross-cluster matchings exceeds $|r_1 - s_1|$. Then by the pigeonhole principle (i.e. via the fact that all points must have exactly one other matched point), there must be at least two such matchings. WLOG, let us say these cross-cluster matches are between $p_i$ and $q_i$, and $p_{i+1}$ and $q_{i+1}$, where $p_i$ and $q_{i+1}$ must be in the same cluster, as are $q_i$ and $p_{i+1}$ since there are only two clusters. By our assumption on the spacing of the clusters, the cost of matching $p_i$ to $q_i$ and $p_{i+1}$ to $q_{i+1}$ at least

$$> 2(2\Delta)^2.$$

However, consider the alternative matching of $p_i$ to $q_{i+1}$ and $p_{i+1}$ to $q_i$. These are both intra-cluster matchings, which by our assumption on the radius of the clusters must have total cost smaller than or equal to

$$\leq 2(2\Delta)^2.$$

This matching has smaller cost than the inter-cluster matching strategy, so this contradicts optimality of the inter-cluster pairing.

In the scenario with $m$ clusters, an analogous argument works. As above, $|r_i - s_i|$ is the number of cluster $i$ elements that must be matched in a cross-cluster fashion. If there are more than the minimum $\frac{1}{2} \sum_i |r_i - s_i|$, by the pigeonhole principle, there must be additional cross-cluster matches that form a cycle in the graph over clusters. As above, the cost would be reduced by matching points within the same clusters, so this contradicts optimality.

Since the number of cross-cluster matchings cannot be less than or exceed $\frac{1}{2} \sum_i |r_i - s_i|$, it must equal this number and the lemma results.

# F   Proof of Theorem 3.4

By Lemma 3.3, if $r_i$ and $s_i$ denote the number of points in cluster $i$ from batch 1 and 2, the resulting number of cross-cluster matches is $\frac{1}{2} \sum_i |r_i - s_i|$. As the samples for our batches are i.i.d., these random variables ($r_i$, $s_i$, where $r_i$ is independent of $s_i$) each (marginally) follow a simple binomial distribution $B(k, p_i)$. We can bound the expectation of this quantity with Jensen's inequality:

$$(\mathbb{E}[|r_i - s_i|])^2 \leq \mathbb{E}[|r_i - s_i|^2] = 2\mathrm{Var}(r_i) = 2kp_i(p_i - 1).$$

This implies that

$$\frac{\frac{1}{2}\sum_i \mathbb{E}|r_i - s_i|}{k} \leq \frac{\sum_i \sqrt{2kp_i(1-p_i)}}{2k} = (2k)^{-\frac{1}{2}}\sum_{i=1}^{m}\sqrt{p_i(1-p_i)},$$

yielding the bound in the theorem. It is also possible to get an exact rate with some hypergeometric identities (Katti, 1960), but these simply differ by a constant factor, so we omit the exact expressions here.

## G  Proof of Theorem 3.5

By the smooth boundary and positive density assumptions, we know that $P(A_\delta) > 0$ and $P(B_\delta) > 0$ for any $\delta > 0$. Hence, for fixed $\delta$ and $k$ large enough, we know that with high probability the sets $A_\delta$ and $B_\delta$ each contain more points than the number of cross-cluster identifications.

Now consider $A_\epsilon$ and $B_\epsilon$ for $\epsilon = 2\delta + (\max(R_A, R_B)^2)/(2D^2)$. All cross-cluster matches need to be assigned. The cost of assigning a cross-cluster match to a point in $A_\delta$ and a point in $B_\delta$ is at most $(1 + 2\delta)^2 D^2$ (since we are using $W_2$). Furthermore, the cost of assigning a cross-cluster match that contains a point in $A$ outside $A_\epsilon$ and an arbitrary point in $B$ is at least $(1 + \epsilon)^2 D^2$. Consider the difference between these two costs:

$$(1+\epsilon)^2 D^2 - (1+2\delta)^2 D^2 = (2(\epsilon - 2\delta) + \epsilon^2 - 4\delta^2)D^2 > 2D^2\frac{\max(R_A, R_B)^2}{2D^2} \geq R_A^2.$$

Since this difference $> 0$ and we have shown $A_\delta$ contains sufficient points for handling all assignments, this assignment outside of $A_\epsilon$ will only occur if there is a within-cluster pair which benefits from using the available point in $A_\epsilon$ more than is lost by not giving it to the cross-cluster pair ($> R_A^2$). The maximum possible benefit gained by the within-cluster pair is the squared radius of $A$, i.e. $R_A^2$. Since we have shown that the lost cost for the cross-cluster pair is bigger than $R_A^2$, we have arrived at a contradiction. The proof is similar for the $B$ side.

We have thus shown that for $k$ large enough (depending on $\delta$), with high probability all cross-cluster matches have an endpoint each in $A_\epsilon$ and $B_\epsilon$ where $\epsilon = 2\delta + (\max(R_A, R_B)^2)/(2D^2)$. Setting $\delta = (\max(R_A, R_B)^2)/(4D^2)$ completes the proof.

## H  Proof of Theorem 4.1

We mostly follow the notation and argument of Zhang et al. (2020) (c.f. Lemma 3.1), modifying it for our setting. There they consider sampling $\lambda \sim Beta(\alpha, \beta)$ from an asymmetric Dirichlet distribution. Here, we assume a symmetric Dirichlet distribution, such that $\alpha = \beta$, simplifying most of the expressions. The analogous results hold in the asymmetric case with simple modifications.

Consider the probability distribution $\tilde{\mathcal{D}}_\lambda$ with probability distribution: $\beta(\alpha+1, \alpha)$. Note that this distribution is more heavily weighted towards 1 across all $\alpha$, and for $\alpha < 1$, there is an asymptote as you approach 1.

Let us adopt the shorthand notation $\tilde{x}_{i,\sigma_{\gamma\zeta}(i)}(\lambda) := \lambda x_i^\gamma + (1-\lambda)x_{\sigma\gamma\zeta}^\zeta$ for an interpolated feature point. The manipulations below are abbreviated, as they do not differ much for our generalization.

$$\mathcal{E}_k^{mix}(f) = \frac{1}{k\binom{N}{k}^2}\mathbb{E}_{\lambda\sim\beta(\alpha,\alpha)}\sum_{\gamma,\zeta=1}^{\binom{N}{k}}\sum_{i=1}^{k}h(f(\tilde{x}_{i,\sigma_{\gamma\zeta}(i)}(\lambda))) - (\lambda y_i^\gamma + (1-\lambda)y_{\sigma_{\gamma\zeta}(i)}^\zeta)f(\tilde{x}_{i,\sigma_{\gamma\zeta}(i)}(\lambda))$$

$$= \frac{1}{k\binom{N}{k}^2}\mathbb{E}_{\lambda\sim\beta(\alpha,\alpha)}\mathbb{E}_{B\sim Bern(\lambda)}\sum_{\gamma,\zeta=1}^{\binom{N}{k}}\sum_{i=1}^{k}B[h(f(\tilde{x}_{i,\sigma_{\gamma\zeta}(i)}(\lambda))) - y_i^\gamma f(\tilde{x}_{i,\sigma_{\gamma\zeta}(i)}(\lambda))]$$

$$+ (1-B)[h(f(\tilde{x}_{i,\sigma_{\gamma\zeta}(i)}(\lambda))) - y_{\sigma\gamma\zeta(i)}^\zeta f(\tilde{x}_{i,\sigma_{\gamma\zeta}(i)}(\lambda))]$$

$$= \frac{1}{k\binom{N}{k}^2}\sum_{\gamma,\zeta=1}^{\binom{N}{k}}\sum_{i=1}^{k}\mathbb{E}_{\lambda\sim\beta(\alpha+1,\alpha)}h(f(\tilde{x}_{i,\sigma_{\gamma\zeta}(i)}(\lambda))) - y_i^\gamma f(\tilde{x}_{i,\sigma_{\gamma\zeta}(i)}(\lambda))$$

For the third equality above, the ordering of sampling for $\lambda$ and $B$ has been swapped via conjugacy: $\lambda \sim \beta(\alpha, \alpha)$, $B|\lambda \sim Bern(\lambda)$ is equivalent to $B \sim \mathcal{U}\{0,1\}$, $\lambda|B \sim \beta(\alpha + B, \alpha + 1 - B)$. This is combined with the fact that $\tilde{x}_{i,\sigma_{\gamma\zeta}(i)}(1 - \lambda) = \tilde{x}_{\sigma_{\gamma\zeta}(i),i}(\lambda)$ to get the last line above.

Now we can swap the sums, grouping over the initial point to express this as the following:

$$\mathcal{E}_k^{mix}(f) = \frac{1}{N} \sum_{i=1}^{N} \mathbb{E}_{\lambda \sim \beta(\alpha+1,\alpha)} \mathbb{E}_{r \sim \mathcal{D}_i} h(f(\lambda x_i + (1 - \lambda)r)) - y_i^{\gamma} f(\lambda x_i + (1 - \lambda)r),$$

where the probability distribution $\mathcal{D}_i$ is as described in the text.

The remainder of the argument performs a Taylor expansion of the loss term $h(f(\lambda x_i + (1 - \lambda)r)) - y_i^{\gamma} f(\lambda x_i + (1 - \lambda)r)$ in terms of $1 - \lambda$, and is not specific to our setting, so we refer the reader to Appendix A.1 of (Zhang et al., 2020). for the argument.

## I $k$-mixup as Mean Reversion followed by Regularization

**Theorem I.1.** *Define $(\tilde{x}_i^{\gamma}, \tilde{y}_i^{\gamma})$ as*

$$\tilde{x}_i^{\gamma} = \bar{x}_i^{\gamma} + \bar{\theta}(x_i^{\gamma} - \bar{x}_i^{\gamma})$$
$$\tilde{y}_i^{\gamma} = \bar{y}_i^{\gamma} + \bar{\theta}(y_i^{\gamma} - \bar{y}_i^{\gamma}),$$

*where $\bar{x}_i^{\gamma} = \frac{1}{\binom{N}{k}} \sum_{\zeta=1}^{\binom{N}{k}} x_{\sigma_{\gamma\zeta}(i)}^{\zeta}$ and $\bar{y}_i^{\gamma} = \frac{1}{\binom{N}{k}} \sum_{\zeta=1}^{\binom{N}{k}} y_{\sigma_{\gamma\zeta}(i)}^{\zeta}$ are expectations under the matchings and $\theta \sim \beta_{[1/2,1]}(\alpha, \alpha)$. Further, denote the zero mean perturbations*

$$\tilde{\delta}_i^{\gamma} = (\theta - \bar{\theta})x_i^{\gamma} + (1 - \theta)x_{\sigma_{\gamma\zeta}(i)}^{\zeta} - (1 - \bar{\theta})\bar{x}_i^{\gamma}$$
$$\tilde{\epsilon}_i^{\gamma} = (\theta - \bar{\theta})y_i^{\gamma} + (1 - \theta)y_{\sigma_{\gamma\zeta}(i)}^{\zeta} - (1 - \bar{\theta})\bar{y}_i^{\gamma}.$$

*Then the k-mixup loss can be written as*

$$\mathcal{E}_k^{OTmixup}(f) = \frac{1}{\binom{N}{k}} \sum_{\gamma=1}^{\binom{N}{k}} \mathbb{E}_{\theta,\zeta} \left[ \frac{1}{k} \sum_{i=1}^{k} \ell(\tilde{y}_i^{\gamma} + \tilde{\epsilon}_i^{\gamma}, f(\tilde{x}_i^{\gamma} + \tilde{\delta}_i^{\gamma})) \right].$$

The mean $\bar{x}_i^{\gamma}$ being shifted toward is exactly the mean of the locally-informed distribution $\mathcal{D}_i$. Moreover, the covariance structure of the perturbations is detailed in the proof (simplified in Section I.1) and is now also derived from the local structure of the distribution, inferred from the optimal transport matchings.

*Proof.* This argument is modelled on a proof of Carratino et al. (2020), so we adopt analogous notation and highlight the differences in our setting and refer the reader to Appendix B.1 of that paper for any omitted details. First, let us use shorthand notation for the interpolated loss function:

$$m_i^{\gamma\zeta}(\lambda) = \ell(f(\lambda x_i^{\gamma} + (1 - \lambda)x_{\sigma_{\gamma\zeta}(i)}^{\zeta}), \lambda y_i^{\gamma} + (1 - \lambda)y_{\sigma_{\gamma\zeta}(i)}^{\zeta}).$$

Then the mixup objective may be written as:

$$\mathcal{E}_k^{mix}(f) = \frac{1}{k\binom{N}{k}^2} \sum_{\gamma,\zeta=1}^{\binom{N}{k}} \sum_{i=1}^{k} \mathbb{E}_{\lambda} m_i^{\gamma\zeta}(\lambda).$$

As $\lambda \sim \beta(\alpha, \alpha)$, we may leverage the symmetry of the sampling function and use a parameter $\theta \sim \beta_{[1/2,1]}(\alpha, \alpha)$ to write the objective as:

$$\mathcal{E}_k^{mix}(f) = \frac{1}{\binom{N}{k}} \sum_{\gamma=1}^{\binom{N}{k}} \ell_i, \qquad \text{where } \ell_i = \frac{1}{k\binom{N}{k}} \sum_{\zeta=1}^{\binom{N}{k}} \sum_{i=1}^{k} \mathbb{E}_{\theta} m_i^{\gamma\zeta}(\theta).$$

To obtain the form of the theorem in the text, we introduce auxiliary variables $\tilde{x}_i^\gamma, \tilde{y}_i^\gamma$ to represent the mean-reverted training points:

$$\tilde{x}_i^\gamma = \mathbb{E}_{\theta,\zeta}\left[\theta x_i^\gamma + (1-\theta)x_{\sigma_{\gamma\zeta}(i)}^\zeta\right]$$
$$\tilde{y}_i^\gamma = \mathbb{E}_{\theta,\zeta}\left[\theta y_i^\gamma + (1-\theta)y_{\sigma_{\gamma\zeta}(i)}^\zeta\right],$$

and $\tilde{\delta}_i^\gamma, \tilde{\epsilon}_i^\gamma$ to denote the zero mean perturbations about these points:

$$\tilde{\delta}_i^\gamma = \theta x_i^\gamma + (1-\theta)x_{\sigma_{\gamma\zeta}(i)}^\zeta - \mathbb{E}_{\theta,\zeta}\left[\theta x_i^\gamma + (1-\theta)x_{\sigma_{\gamma\zeta}(i)}^\zeta\right]$$
$$\tilde{\epsilon}_i^\gamma = \theta y_i^\gamma + (1-\theta)y_{\sigma_{\gamma\zeta}(i)}^\zeta - \mathbb{E}_{\theta,\zeta}\left[\theta y_i^\gamma + (1-\theta)y_{\sigma_{\gamma\zeta}(i)}^\zeta\right].$$

These reduce to the simplified expressions given in the theorem if we recall that $\theta$ and $\zeta$ are independent random variables. Note that both the mean-reverted points and the perturbations are informed by the local distribution $\mathcal{D}_i$. $\qquad\square$

## I.1 Covariance structure

As in (Carratino et al., 2020), it is possible to come up with some simple expressions for the covariance structure of the local perturbations, hence we write out the analogous result below. As the argument is very similar to that in (Carratino et al., 2020), we omit it.

**Lemma I.2.** *Let $\sigma^2$ denote the variance of $\beta_{[1/2,1]}(\alpha,\alpha)$, and $\nu^2 := \sigma^2 + (1-\bar{\theta})^2$. Then the following expressions hold for the covariance of the zero mean perturbations:*

$$\mathbb{E}_{\theta,\zeta}\tilde{\delta}_i^\gamma(\tilde{\delta}_i^\gamma)^\top = \frac{\sigma^2(\tilde{x}_i^\gamma - \bar{x}_i^\gamma)(\tilde{x}_i^\gamma - \bar{x}_i^\gamma)^\top + \nu^2\Sigma_{\tilde{x}_i^\gamma\tilde{x}_i^\gamma}}{\bar{\theta}^2}$$

$$\mathbb{E}_{\theta,\zeta}\tilde{\epsilon}_i^\gamma(\tilde{\epsilon}_i^\gamma)^\top = \frac{\sigma^2(\tilde{y}_i^\gamma - \bar{y}_i^\gamma)(\tilde{y}_i^\gamma - \bar{y}_i^\gamma)^\top + \nu^2\Sigma_{\tilde{y}_i^\gamma\tilde{y}_i^\gamma}}{\bar{\theta}^2}$$

$$\mathbb{E}_{\theta,\zeta}\tilde{\delta}_i^\gamma(\tilde{\epsilon}_i^\gamma)^\top = \frac{\sigma^2(\tilde{x}_i^\gamma - \bar{x}_i^\gamma)(\tilde{y}_i^\gamma - \bar{y}_i^\gamma)^\top + \nu^2\Sigma_{\tilde{x}_i^\gamma\tilde{y}_i^\gamma}}{\bar{\theta}^2},$$

*where $\Sigma_{\tilde{x}_i^\gamma\tilde{x}_i^\gamma}, \Sigma_{\tilde{y}_i^\gamma\tilde{y}_i^\gamma}, \Sigma_{\tilde{x}_i^\gamma\tilde{y}_i^\gamma}$ denote empirical covariance matrices.*

Note again, that the covariances above are locally-informed, rather than globally determined. Lastly, there is also a quadratic expansion performed about the mean-reverted points $\tilde{x}_i^\gamma, \tilde{y}_i^\gamma$ with terms that regularize $f$, but we omit this result as the regularization of Theorem 4.1 is more intuitive (c.f. Theorem 2 of (Carratino et al., 2020)).

## J  Vicinal Distribution of $k$ Nearest-Neighbors

We provide this section for explicit illustration of why $k$ nearest-neighbors is not a sensible alternative for $k$-mixup. In this setting, we consider drawing two sets of $k$ samples $\{(x_i^\gamma, y_i^\gamma)\}_{i=1}^k$ and $\{(x_i^\zeta, y_i^\zeta)\}_{i=1}^k$, match each $x_i^\gamma$ with its nearest-neighbor in $\{x_i^\zeta\}_{i=1}^k$, and then interpolate these matches to obtain the vicinal distribution. In Figure 7, we provide example images of the generated matching distributions. Note that in general, these matching distributions have very limited cross-cluster matchings. The "Swiss Roll" dataset is an exception due to the interwoven arms of the spirals. Additionally, note that the intra-cluster matchings are much more concentrated around the points in question, and do not perform as much smoothing within clusters.

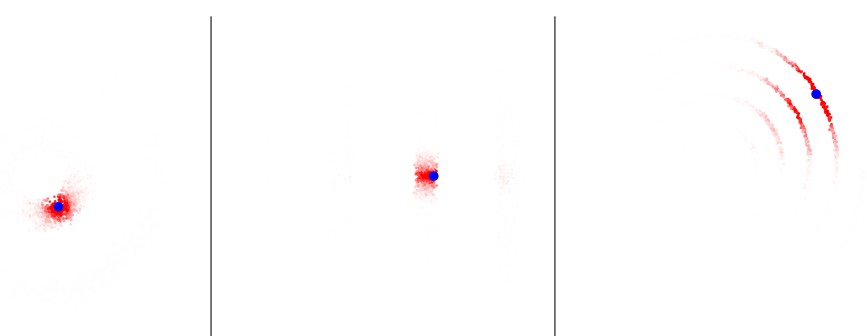

Figure 7: Example of matching distributions generated by $k = 32$-nearest-neighbors, using the same point as in Figure 3.

