# OpenReview forum: "$k$-Mixup Regularization for Deep Learning via Optimal Transport"
_TMLR — Accepted by TMLR_

### Review · Reviewer_6SVz · 2023-06-06

**Summary Of Contributions:**

This paper proposes k-Mixup which is a data augmentation technique mixing k (k is generally bigger than 1) instances based on the optimal transport. The authors theoretically and empirically demonstrate that k-mixup is more likely to generate instances within the data manifold compared with 1-mixup. For data of several clusters, k-mixup generates a higher proportion of within-cluster instances instead of inter-cluster ones with the increase of k. Extensive experiments in various applications demonstrate the effectiveness of k-mixup and its advantages over 1-mixup. In addition, the authors show models trained by k-mixup demonstrate robustness against random noise or simple FGSM perturbation.

**Audience:**

Yes

**Broader Impact Concerns:**

This paper proposed a generic method and there is no concerns on ethics as far as what I know.

**Claims And Evidence:**

Yes

**Requested Changes:**

Generally, this work is well written, well motivated and easy to follow. The revision should address the weakness part pointed in the previous section.

**Strengths And Weaknesses:**

Strength:

1. The method is well motivated and justified theoretically and empirically.
2. The experiments are comprehensive and in different domains / applications.

Weakness:

1. [Proposition 3.2] This proposition indicates that the instances generated by k-mixup will be in the manifold with high probability. However, is this enough to justify the advantages of k-mixup? It is sort of necessary condition instead of a sufficient condition, which should be like: can the data in the manifold be generated by k-mixup? Since even for 1-mixup, the generated data would be highly likely in the manifold when $\alpha$ is very small.

2. For the experiments, Table 1, since every setting is run for 20 times, it is better to report the performance variance.

3. In Figure 5, what is the definition of "train accuracy"? Since the generate data does not have a one-hot label.

4. Based on the motivation and the theoretical analysis, the algorithm should work better with a bigger $k$, but this is not always true based on the results in Table 3. Can you explain the performance degradation when $k$ is too large?

---

> ### Author Response · Authors · 2023-06-30
> **Response**
>
> Thank you for noting that the method is well-motivated and justified both theoretically and empirically. We were happy to see that this opinion was generally shared amongst the reviewers.
>
> **Prop 3.2:** The key here is showing that the matches will lie along the manifold, so that the sampled points will be in-manifold regardless of the choice of $\alpha$. While it is true that very small $\alpha$ will ensure that 1-mixup perturbations stay close to the data manifold, this will be entirely by accident since the direction of sampling does not lie along the manifold and may be orthogonal to it. Hence for 1-mixup, low-alpha samples will fail to sample *along* the manifold as mentioned by the reviewer, and the resulting perturbations may be too small to result in enough regularization for improved performance. This can be seen in the experiments, where small $\alpha$ usually does not produce optimal performance. $k$-mixup is more robust to larger $\alpha$, and is capable of motivating within-manifold perturbations even when $\alpha$ is large. The results of this can be seen clearly in nearly every experiment we ran, by looking at the highest $\alpha$ considered, and seeing the monotonic increase in performance with increasing $k$ for that column.
>
> **Performance variance:** We provide confidence intervals in later results tables, but we will add the performance variance in our revision.
>
> **Figure 5:** This is a good point, for train samples involving interpolated class labels, in order to obtain a percent accuracy value, we round towards the dominant label and use this as the “true label” of the vicinal sample. We do not present this metric as necessarily being a good measure to evaluate the performance of the model, but rather as potentially of interest in observing training dynamics.
>
> **Large $k$:** Firstly, note that, for instance, the best $k$ for each of the image dataset experiments are $k=64$, $k=16$, $k=16$, $k=16$, and $k=16$, i.e. the best $k$ is consistently one of the largest $k$'s tested. Overall, we find that having moderate alpha ($\approx 1$), and relatively high $k$ results in strong performance over several datasets and network architectures. Secondly, note that the theory is not directly analyzing test error. Hence, we don't necessarily have a direct theoretical reason to expect that increasing $k$ without limit would always monotonically improve test classification performance, we simply show that generally speaking large $k$ provides good regularization properties that should be beneficial (as is in fact observed in the experiments). Finally, real datasets are always messy, it would be extremely surprising if we saw monotonic test classification error improvement with increasing $k$ across so many real datasets.
>
> That said, there are two interesting experimental regimes where we *do* tend to more explicitly see clearly improving performance with increasing $k$.
>
> The first such regime is the one for fixed, large perturbation size $\xi$. Here, increasing $k$ clearly increases performance (almost) monotonically. For lower values of $\xi$, the magnitude of the perturbations is much smaller, so the quality of the matchings likely does not have as much of an effect on the vicinal distribution. As a result, the generalization improvements become more gradual, with many of the performance "non-monotonicities" in $k$ here being on the order of the error bars of the Monte Carlo trial. An intuitive explanation of this phenomenon is that when perturbation distances are large, we expect that the importance of having well-informed matchings will increase.
>
> The second such regime is the setting of Tables 6 and 7, where we are testing performance under distribution shift. Here, there is a clear trend of the largest $k$ performing best, with diminishing returns for increasing $k$ further being the primary reason not to use even larger $k$. Intuitively, this trend makes sense as good performance in the distribution shift setting is highly dependent on high-quality regularization, meaning that the improving properties predicted by our theory are more impactful in this performance metric.

---

### Review · Reviewer_tNWs · 2023-06-16

**Summary Of Contributions:**

This paper proposes an extension of the popular Mixup regularization technique. In particular, instead of simply taking convex combinations of random data points, this work proposes interpolating between k-batches of data, each representing a discrete distribution in itself, where interpolation is performed under the Wasserstein metric. Theoretical results indicate that the proposed k-Mixup technique better preserve the original structure of the data compared to standard Mixup. Numerical experiments on various datasets across different modalities suggest that k-Mixup can further improve generalization and robustness to some types of perturbations.

**Audience:**

Yes

**Broader Impact Concerns:**

No concerns.

**Claims And Evidence:**

Yes

**Requested Changes:**

I think the following additions would strengthen the work's practical contribution significantly:
- Additional experiments on large-scale vision datasets (see details under Weaknesses above).
- More rigorous comparison of the compute requirements of Mixup/k-Mixup, potentially moving the analysis to the main paper from the Appendix.

Furthermore, I think it would improve the readability of the paper if the Algorithm summary/implementation from the appendix was moved to the main paper.

**Strengths And Weaknesses:**

Strengths:
- In my opinion, the paper is well-written and easy to follow in most parts. More involved technical concepts, are illustrated nicely through figures that help conveying the intuition of the paper.
- The theoretical contributions on preservation of global structure and locally-informed regularization looks solid.
- Empirical results consistently demonstrate that generalization using k-Mixup is not worse than standard Mixup after tuning a hyperparameter.
- Based on the experiments, the generalization improvement over standard Mixup is significant in the presence of additive Gaussian perturbations with large variance.

Weaknesses:
- Experiments are on datasets that can be considered small/low-dimensional with today's standard. It would be interesting to see how the method performs on ImageNet with a larger ResNet model for instance. As the improvements over standard Mixup are fairly small in some cases, it would be important to demonstrate that the computational overhead introduced by k-Mixup on larger datasets is worthwhile. Otherwise, the practical contribution of the paper is somewhat limited to small datasets.
- I believe the compute cost analysis should be better highlighted in the paper and more rigorously compared to standard Mixup, as this is the main trade-off when deciding between Mixup and k-Mixup.

---

> ### Author Response · Authors · 2023-06-30
> **Response**
>
> Thank you for noting that the method is well-motivated and justified both theoretically and empirically. We were happy to see that this opinion was generally shared amongst the reviewers.
>
> **Compute cost analysis:** We are happy to report that the additional computational cost from using $k$-mixup (at least for the size of $k$ we use in the paper) is virtually nil relative to the baseline cost of training the network without mixup. We have moved a discussion of this fact from the appendix to the main text in Section 5, as we agree this is an important point.
>
> **Algorithm summary location:** As suggested, we have moved this to Section 5 in the main text (as long as the AE finds the increased main text length acceptable).
>
> **ImageNet:** We agree this would be an interesting experiment. Unfortunately, for reasons beyond our best efforts to address, this type of experiment is beyond the computational resources of our lab. This is particularly true as several dozen Monte Carlo runs of training would be required to ensure that our observed results were statistically significant. Complicating things further is the fact that our industry collaborator is not permitted by their employer to use their computational resources on the full ImageNet dataset due to significant legal and ethical concerns regarding certain images in the dataset. We hope that the reviewer can be satisfied by our extensive experiments on smaller datasets across a variety of domains, and the promising experiments on the TinyImageNet subset of classic ImageNet. We point out that neither our empirical nor theoretical results indicate performance issues with increasing scale, hence we have every reason to believe that $k$-mixup would be similarly beneficial on ImageNet.

---

### Review · Reviewer_dnUq · 2023-06-25

**Summary Of Contributions:**

The major contribution of this paper is that it proposes a new mixup method based on optimal transport, which can lead to improved generalization performance than the standard Mixup. Detailed contributions are summarized as follows:

* The proposed method can give better empirical generalization performance.
* This paper develops a theoretical analysis to justify that the proposed methods can generate interpolated samples that are closer to the data manifold.
* This paper also theoretically shows that the proposed k-mixup method can lead to more informed regularizations.


**Audience:**

Yes

**Broader Impact Concerns:**

No concern

**Claims And Evidence:**

Yes

**Requested Changes:**

See the weakness section.

**Strengths And Weaknesses:**

Strength:

* The proposed method is theoretically sound and achieves better empirical performance than baselines.

Weakness:

* The empirical performance improvement is not that significant and it requires an additional hyperparameter $k$.
* The authors may also need to provide the run-time analysis as it is not clear whether calculating the optimal match is time-consuming or not.
* The motivation that the data point generalized by Mixup should lie in the data manifold needs more justifications. For instance, the authors should also consider performing Mixup for the data pair with the same labels (as the data generated by this type of mixup could be closer to the data manifold than the data generated from two data points with different labels.)

---

> ### Author Response · Authors · 2023-06-30
> **Response**
>
> Thank you for noting that the method is well-motivated and justified both theoretically and empirically. We were happy to see that this opinion was generally shared amongst the reviewers.
>
> **Computation and Hyperparameter** In response to the first two points above (and to a request by Reviewer tNWs), we have moved Supplement Section A to the main text.
>
> We acknowledge the additional hyperparameter, and provide several strategies for parameter tuning in Supplement Section A. As noted there, using $\alpha = 1$ and optimizing over $k$ being small powers of 2, outperforms 1-mixup in nearly every case. Our experiments show that the performance gains of k-mixup (with tuning) are comparable to those of mixup over ERM, so we feel that the additional computational cost of tuning is not too great for the improvement it results in.
>
> The runtime analysis is also provided in Supplement Section A. The amortized cost (per datapoint) of OT matching and the cost matrix computation is $O(k)$ and $O(kd)$, respectively (where $d$ is the dataset dimension). Ultimately, with $k < 50$ as we’ve used in our experiments, the OT matching cost is very small relative to total epoch cost. In the case of $k=32$, with a high-dimensional dataset like CIFAR, the cost of matching is 0.69 seconds, and the cost matrix computation is approximately 0.5 seconds. The overall epoch time is more than 30 seconds.
>
> **Labels** As implemented, k-Mixup *will* produce matchings both between datapoints of the same label and between datapoints of different labels, as it selects random pairs of k-batches, regardless of their label. The resulting interpolation within classes is important for augmenting intra-class data, and will contribute to model regularization, as the cross-class interpolations do. Overall, these interpolations are more and more likely to remain within the data manifold as $k$ increases, ensuring more globally-informed perturbations. Regarding the suggestion by the reviewer - we did try label-aware matching schemes, both (a) only allowing same-class matches and (b) only allowing different-class matches. Both approaches performed worse than our presented approach. Based on the fact that our proposed label-agnostic approach corresponds to the label-agnostic approach of the original mixup paper, and we cannot find a good intuitive motivation for either of the (a) or (b) label-aware approaches, we have opted to omit this negative result from the paper.

---

### Decision · Action_Editors · 2023-08-15

**Recommendation:** Accept as is

**Comment:**

The reviewers all found the method was well-motivated and the experiments were correct. There were some concerns about the lack of large-scale experiments, but the experiments provided were comprehensive and technically correct. Therefore, I recommend acceptance.

**Audience:**

The development and understanding of regularization methods for deep networks is an open area of research and is of interest to the TMLR community.

**Claims And Evidence:**

This paper proposes a novel method inspired by Mixup based on optimal transport. A key claim is that the proposed method can generate interpolated samples that are closer to the data manifold. The reviewers found that the paper is theoretically sound and the method is evaluated on a variety of datasets.